# Environmental stiffness regulates neuronal maturation via Piezo1-mediated transthyretin activity

Eva Kreysing [1,2,3,4] ✉, Hélène O. B. Gautier[1], Sudipta Mukherjee [1,2,3], Katrin A. Mooslehner[1,2,3], Leila Muresan [1,5], Daniel Haarhoff[6], Xiaohui Zhao [7], Alexander K. Winkel[1], Tina Borić [2,3], Sebastián Vásquez-Sepúlveda[2,3], Niklas Gampl [2,3], Andrea Dimitracopoulos [1], Eva K. Pillai [1], Robert Humphrey[1,8], Ragnhildur Thóra Káradóttir [8] ✉ & Kristian Franze [1,2,3] ✉

During development, neurons initiate a maturation process during which they start expressing voltage-gated ion channels, form synapses, and start communicating via action potentials. Little is known about external factors regulating this process. Here, we identify environmental mechanics as an important regulator of neuronal maturation, and a molecular pathway linking tissue stiffness to this process. Using patch clamp electrophysiology, calcium imaging and immunofluorescence, we find that, in stiffer environments, neurons show a delay in voltage-gated ion channel activity, action potentials, and synapse formation. RNA sequencing and CRISPR/Cas9 knockdown reveal that the mechanosensitive ion channel Piezo1 supresses transthyretin expression on stiffer substrates, slowing down electrical maturation. In *Xenopus laevis* embryos, brain stiffness negatively correlates with synapse density, and artificial tissue stiffening delays synaptic activity in vivo. Our data indicate that environmental stiffness represents a fundamental regulator of neuronal maturation, critical for brain circuit development and potentially for neurodevelopmental disorders.

Cell maturation is a fundamental aspect of the development of multicellular organisms, underlying cellular specialisation in many different cell types. During this process, cells become fully differentiated, attaining specialised features and functions. Hallmarks of cell maturation include a spectrum of changes in proliferation dynamics, gene expression profiles, metabolic activity, morphology, cellular connectivity and functionality[1].

In the nervous system, the maturation of postmitotic neurons is essential for neurons to sense, process, transmit and store information. During the maturation process, neurons extend axons and dendrites, become electrically active, form synapses, express transmitters and transmitter receptors, and eventually form complex networks[2]. These processes are regulated mostly by genetic and epigenetic factors[3].

Some of these processes are also regulated by external factors though[3]. For example, axon outgrowth and neurotransmitter expression[4] are promoted by brain-derived neurotrophic factor (BDNF), the development of presynaptic specialisation sites in axons is regulated

[1]Department of Physiology, Development and Neuroscience, University of Cambridge, Cambridge, UK. [2]Medical Institute of Biophysics, Friedrich-Alexander-Universität Erlangen-Nürnberg, Erlangen, Germany. [3]Max-Planck-Zentrum für Physik und Medizin, Erlangen, Germany. [4]Warwick Medical School, University of Warwick, Coventry, UK. [5]School of Computing and Information Science, Faculty of Science and Engineering, Anglia Ruskin University, Cambridge, UK. [6]Makespace Cambridge Ltd, Cambridge, UK. [7]Department of Medicine, Cardiovascular Division, University of Cambridge, Heart and Lung Research Institute, Cambridge, UK. [8]Cambridge Stem Cell Institute, Jeffrey Cheah Biomedical Centre, University of Cambridge Biomedical Campus, Cambridge, UK. ✉e-mail: eva.kreysing@warwick.ac.uk; rk385@cam.ac.uk; kf284@cam.ac.uk

by netrins[5], and synapse formation is promoted by Eph/ephrin signalling through N-methyl-D-aspartate receptor (NMDAR) clustering[6]. However, the electrical maturation of neurons has been thought to be mainly cell-intrinsically regulated.

During electrical maturation, neurotransmitters, neurotransmitter receptors, and voltage-gated ion channels are expressed, enabling neuronal communication via action potentials[7]. Electrical maturation rates are activity-dependent. For example, calcium influxes through voltage-gated ion channels regulate the differentiation of potassium current kinetics at early developmental stages[8]. Furthermore, in the human cortex, the timing of neuronal maturation is regulated by epigenetic factors such as histone methylation, and the rate at which human neurons mature was recently suggested to be determined well before neurogenesis[9]. Yet, the electrical excitability of neurons does not start simultaneously throughout the brain[10], suggesting that external signals may also exist which regulate the timing of the electrical maturation of neurons. Understanding how environmental cues regulate neuronal maturation remains a major challenge in the field[11].

In other tissues, such as cardiac tissue, mechanical stimulation enhances cell maturation[12]. Also in the CNS, tissue mechanics has been shown to instruct various neurodevelopment processes[13], including axon growth and guidance[14,15], dendrite branching[16,17] and the activity of established neuronal networks[18]. However, the precise regulation of synapse formation and electrical maturation remains unknown. Considering the mechanical heterogeneity of the developing nervous system[13], we here explored how mechanical signals regulate neuronal maturation. We found a strong dependence of the temporal development of the electrical maturity of neurons on the stiffness of their environment in vitro as well as in vivo, and identified transthyretin (TTR), which regulates synaptogenesis and neuronal maturation, as a downstream effector of the mechanosensitive ion channel Piezo1.

## Results

### Synaptic density is lower on stiff substrates compared to soft ones

To investigate the effect of substrate stiffness on synapse formation, we cultured primary hippocampal neurons from E17 or E18 rat embryos on soft and stiff polyacrylamide gels coated with Poly-D-Lysine (PDL) and laminin. The shear elastic modulus of soft substrates was ~0.1 kPa and that of stiff substrates ~10 kPa, corresponding to in vivo measurements of soft and stiff brain regions, respectively[19]. After 10 and 14 days of culture (DIV 10 and DIV 14), the cells were fixed and stained for the presynaptic markers vesicular glutamate transporters (VGLUT2) and vesicular GABA transporters (VGAT), the

postsynaptic cell adhesion protein neuroligin (NRL), and neurofilaments (NF), which are particularly abundant in axons. Cells were then imaged using confocal laser scanning microscopy (Fig. 1a, b) and analysed with a custom synapse density analysis script (see "Methods" section). Synapses were identified if a pre- and postsynaptic marker co-localised within 2 pixels (360 nm) of each other and the neurofilament marker (Supplementary Fig. 1). Synapse densities were determined by dividing the synapse counts by the total length of neurofilaments in each field of view.

The density of both inhibitory GABAergic and excitatory glutamatergic synapses was significantly lower on stiff substrates than on soft substrates at both DIV 10 (Fig. 1c) and at DIV 14 (Fig. 1d). These results suggested that synapse formation, which is a prerequisite for the establishment of functional neuronal circuits, is more efficient on soft substrates.

To test whether substrate stiffness-dependent differences in synaptic densities could be attributed to variations in neuronal growth and arborisation (i.e., neurite branching), we conducted a Sholl analysis[20]. Greater arbour complexity could increase the number of contact points between neurons and thereby promote synapse formation. However, the analysis revealed no significant differences in arbour complexity between the different conditions (Supplementary Fig. 2), in line with previously published data[21], suggesting that not neurite growth but a different—substrate stiffness-dependent—mechanism accounted for the observed differences in synaptic densities.

### Sodium current density is lower on stiff substrates compared to soft ones

The presence of voltage-gated ion channels in the cell membrane is essential for the proper function of mature neuronal circuits. These ion channels contribute to a neuron's excitability, i.e., the likelihood that neurons will fire action potentials (APs) in response to stimuli. To explore if substrate stiffness impacted the timing of neuronal electrical excitability, we measured current density - a measure of the electric current flow across the cell membrane - on substrates of different stiffness. Current densities were calculated by normalising the measured currents by the membrane capacitance, which is proportional to the surface area. We focused on three ion channel currents: voltage-gated sodium currents ($I_{Na}$), voltage-gated potassium currents ($I_K$), and delayed rectifier potassium currents ($I_{Kdr}$). Current densities were assessed using whole-cell patch-clamp recordings (see "Methods" section) on primary hippocampal neurons at multiple time points during the maturation period up to DIV 14.

The capacitance of the cells increased throughout the duration of the 14-day culture period (between DIV 2 and 14 by a factor of ~4,

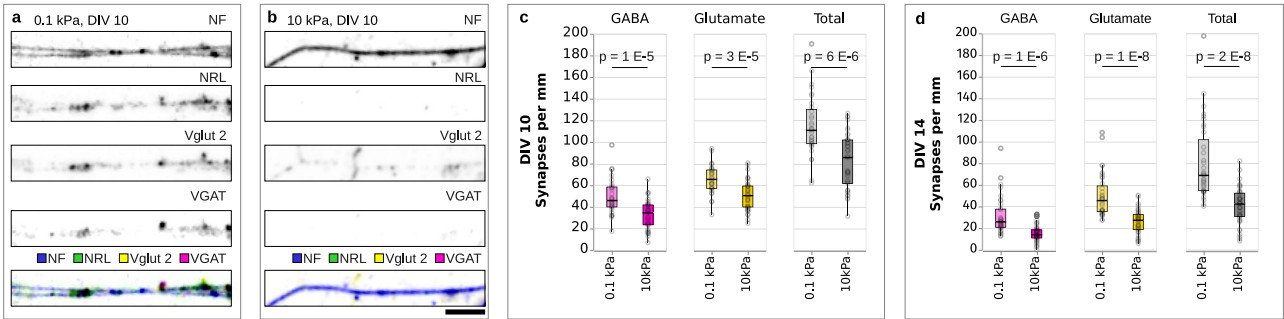

**Fig. 1 | Synapse densities are lower on stiff substrates than on soft substrates.** Immunocytochemistry of neurons cultured on **a** soft (0.1 kPa) and **b** stiff (10 kPa) substrates, showing (from top to bottom) neurofilament (NF), postsynaptic marker (NRL), vesicular glutamate transporter 2 (Vglut2; excitatory), vesicular GABA transporter (VGAT; inhibitory), and an overlay of all channels. **c, d** Quantification of synapse densities. Both glutamatergic and GABAergic synapse densities were lower on stiff substrates than on soft ones at DIVs 10 and 14. Boxplots showing detected synapses per mm neurofilament; *p*-values provided above the plots (two-tailed t-tests). Data from three independent experiments. For DIV 10, 31 fields of view (FOV) were analysed on 0.1 kPa substrates and 37 FOV on 10 kPa substrates. For DIV 14, 35 FOV were analysed on 0.1 kPa substrates and 40 FOV on 10 kPa substrates. Boxplots show the median (central line), the interquartile range (boxes), and whiskers represent 1.5 times the interquartile range. Scale bar: 5 µm.

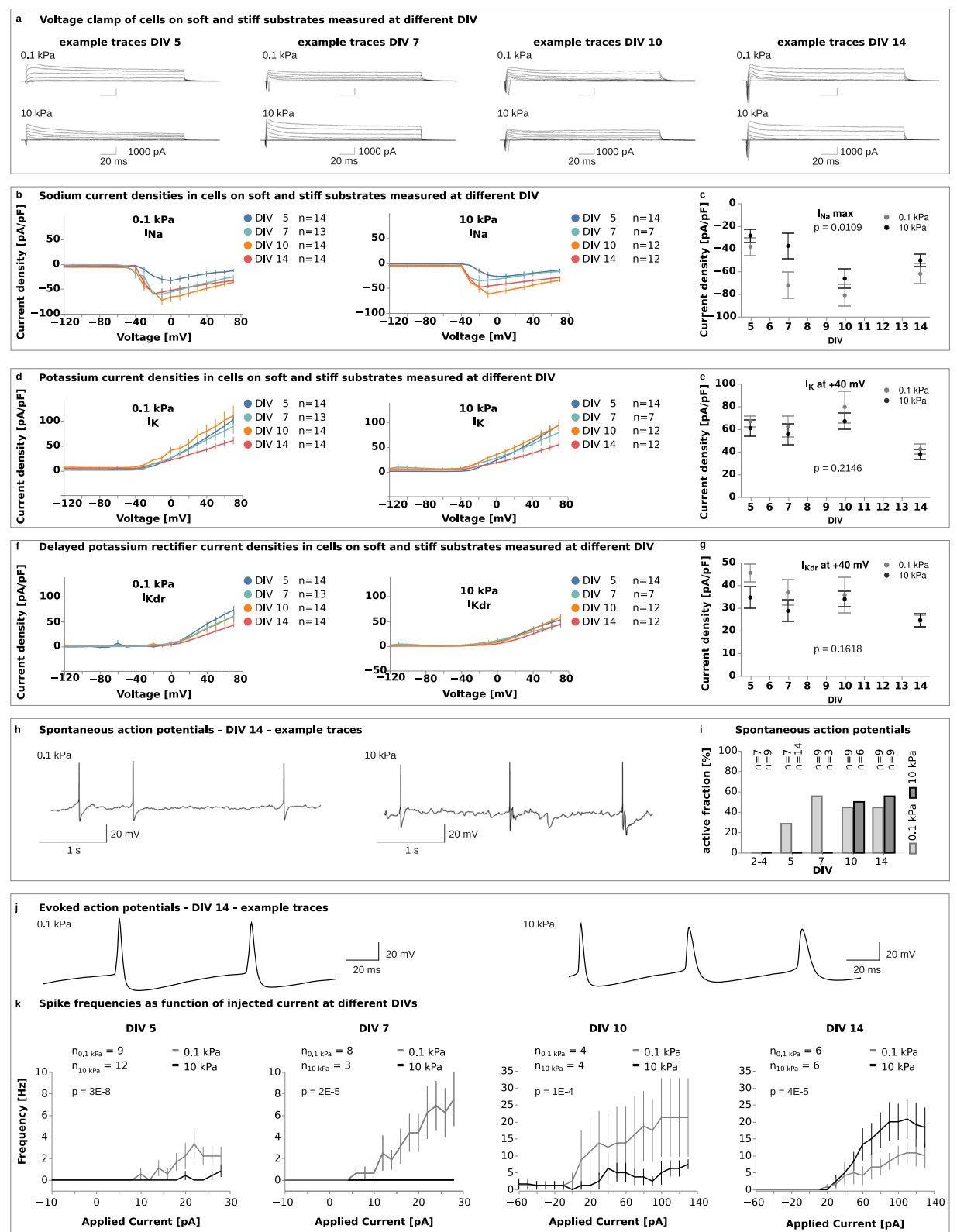

Supplementary Fig. 3), indicating that the neurons kept growing throughout the culture period. Increases in capacitance were similar on both types of substrates (Supplementary Fig. 3), confirming that hippocampal neuronal growth was independent of substrate stiffness (cf. Supplementary Fig. 2).

Cells cultured on stiff substrates showed significantly lower peak sodium current densities $I_{Na\ max}$ compared to those on soft substrates

(Fig. 2b, c), suggesting that voltage-gated sodium channels were expressed earlier and maturation happened sooner in neurons grown on softer substrates. When assessing the $I_K$ and $I_{Kdr}$ densities at +40 mV, we found trends towards lower current densities on stiff substrates; these differences were not statistically significant though (Fig. 2d–g). We also observed a gradual decrease in membrane resistance from DIV 5 to DIV 14, which is expected over the course of

**Fig. 2 | Neuronal activity is lower on stiff substrates than on soft substrates.**
**a** Example traces from voltage-clamp recordings showing induced currents in cells on soft (0.1 kPa) and stiff (10 kPa) substrates at DIV 5, 7, 10, and 14. **b, c** Current densities for voltage-gated sodium (INa) channels were calculated from whole-cell voltage-clamp recordings, as shown in (**a**). **c** Peak sodium current densities were significantly lower on stiff substrates, indicating delayed maturation (two-way ANOVA, time points and substrate stiffness as factors). Cell numbers are identical in (**b**) and (**c**). Current densities for voltage-gated potassium channels (**d, e** IK, **f, g** IKdr) calculated from whole-cell voltage-clamp recordings, as shown in (**a**). **e, g** Current densities at +40 mV were similar between the stiffnesses at all time points (two-way ANOVA). Cell numbers are identical in (**d–g**). **h** Representative traces of spontaneous action potentials (APs) recorded by whole-cell patch-clamp electrophysiology at DIV 14. **i** Plot of spontaneous activity. A cell was considered active if it produced at least one action potential during the recording. On soft gels, spontaneous APs were detected from DIV 5 until the end of the culture period, whilst neurons on stiff gels did not show any spontaneous activity until DIV 10. At DIVs 10 and 14, cells showed similar levels of activity regardless of the environmental stiffness, suggesting that the electrical maturation of neurons was delayed on stiffer substrates. **j** Representative traces of APs evoked in whole-cell current-clamp recordings at DIV 14. **k** F−I curves showing AP frequencies as a function of the injected current. Neuronal excitability on soft and stiff substrates was evaluated using a two-way ANOVA, with applied current and substrate stiffness as factors. Excitability was significantly higher on soft substrates than on stiff ones between DIV 5–10; this trend reversed at DIV 14. **b, d, f, k** Line plots represent mean values, error bars the standard error of the mean (SEM). **c, e, g** Plots represent means, error bars the SEM.

---

maturation[22] (Supplementary Fig. 4a). In contrast, the resting membrane potential remained relatively stable over time (Supplementary Fig. 4b). Neither membrane resistance nor resting membrane potential depended on substrate stiffness (Supplementary Fig. 4).

## Spontaneous and evoked action potentials are delayed on stiff substrates

Neurons communicate via action potentials, which are rapid changes in the voltage across their membrane. The initial rapid rise in potential ('depolarisation') is initiated by the opening of sodium ion channels in the plasma membrane. Since the voltage-gated sodium current density was lower in neurons cultured on stiff substrates, substrate stiffness could potentially affect the generation of action potentials and the formation of functional neuronal circuits.

To test this hypothesis, we investigated whether substrate stiffness affected the ability of neurons to generate spontaneous and evoked action potentials over a 14-day culture period. Spontaneous action potentials occur in neurons in the absence of external stimuli and are commonly observed during healthy neuronal and network maturation[23]. Evoked action potentials, on the other hand, are generated in response to experimental stimuli, typically through current injection.

Neurons grown on soft substrates began generating spontaneous action potentials as early as DIV 5, whereas neurons on stiff substrates only started spontaneous firing at DIV 10 (Fig. 2i). These results indicated that the onset of spontaneous action potential activity was delayed in neurons grown on stiffer substrates, and neurons on softer substrates matured more rapidly. By DIV 10, neurons on stiff substrates caught up and showed a comparable proportion of electrically active cells (~50%, Fig. 2i). A logistic regression analysis revealed a delay in the onset of spontaneous action potentials between neurons cultured on soft and stiff substrates of approximately 4 days (Supplementary Fig. 5).

We observed a similar delay in action potential (AP) generation when using current injections. AP frequencies across current steps were summarised using frequency−current (F−I) curves. From DIV 5 to DIV 10, the F−I curves on stiff substrates consistently lay below those on soft substrates, indicating lower excitability on stiff substrates (Fig. 2k). By DIV 14, this pattern reversed, with neurons on stiff substrates showing higher excitability. Together with our data on synapse formation (Fig. 1a−d), sodium current densities (Fig. 2b, c), and spontaneous neuronal activity (Fig. 2i), these data indicated a delay in neuronal maturation on stiff substrates.

## Substrate stiffness-dependence of neuronal activity is mediated by Piezo1

We next investigated molecular mechanisms underlying the neuronal responses to substrate stiffness ('mechanotransduction'). Since the mechanosensitive ion channel, Piezo1, is expressed in neurons[14,24] and activated in a substrate stiffness-dependent manner at least in fibroblasts[25], we established two independent CRISPR-Cas9 assays to downregulate the expression of Piezo1 (P1) in primary hippocampal neurons. Each knockdown (KD) condition (P1 KD1 and P1 KD2) was modified with 4 different guides designed to target different regions in the *Piezo1* gene, while the control (CTRL) was treated with Cas9 protein and CRISPR RNA only ("Methods" section, Supplementary Fig. 6–8).

To assess the impact of Piezo1 on current densities and neuronal excitability, we measured these parameters at DIV 7 − when differences between soft and stiff substrates were most pronounced (Fig. 2c, i) − in P1 KD (using the P1 KD2 condition, see "Methods" section) and control (CTRL) cells cultured on both substrate types (Fig. 3a–d, and Supplementary Fig. 9). The capacitance of the cells was independent of substrate stiffness and Piezo1 expression (Supplementary Fig. 10).

On soft substrates, sodium current densities in CTRL and P1 KD cells largely overlapped, whereas on stiff substrates, the peak current appeared lower in CTRL cells compared to P1 KD cells (Fig. 3b). Although this difference was visually apparent in the plot, a two-way ANOVA did not yield statistical significance.

We then tested the excitability of CTRL and P1 KD neurons on soft and stiff substrates using current injections. While both groups showed similar excitability on soft substrates, CTRL neurons exhibited significantly lower excitability on stiff substrates compared to P1 KD cells (Fig. 3d), suggesting that the electrical maturation of neurons was delayed on stiff substrates through activation of Piezo1.

Next, we investigated functional consequences of Piezo1 knockdown on spontaneous neuronal activity on soft and stiff substrates at DIV 5, 6, and 7. To increase throughput, we switched to calcium imaging, which reliably measures action potentials[26]. This approach allowed us to include a second Piezo1 knockdown condition, which allowed us to exclude off-target effects. Data were analysed with a custom-written automated script that detected calcium transients with a sensitivity of 96% and a specificity of 98% (Fig. 3e–g, Supplementary Figs. 11, 12; see "Methods" section for details).

At DIV 5, we detected very little activity across all conditions. Consistent with our patch-clamp experiments (Fig. 2i), from DIV 6 onwards, the fraction of active Piezo1 control cells (see "Methods" section for details) was significantly higher on soft substrates than on stiff ones (Fig. 3h, i, k; Supplementary Movies 1, 2), even if lower than in wild-type cells on comparable substrates. In fact, while the fraction of active cells increased steadily on soft substrates, barely any cells showed any activity on stiff substrates. Inhibition of fast voltage-gated sodium channels and thus action potentials with tetrodotoxin (TTX) resulted in a suppression of calcium signals, indicating that the observed calcium peaks were specific to neuronal electrical activity (Supplementary Fig. 13).

Knockdown of Piezo1 led to a significant increase in the activity of neurons cultured on stiff substrates starting from DIV 6 (Fig. 3j, k; Supplementary Movies 3, 4). The activity of neurons cultured on soft substrates, on the other hand, was not significantly altered by the Piezo1 knockdown. The fraction of KD cells showing calcium transients on stiff substrates was similar to that of CTRL neurons cultured on soft substrates (Supplementary

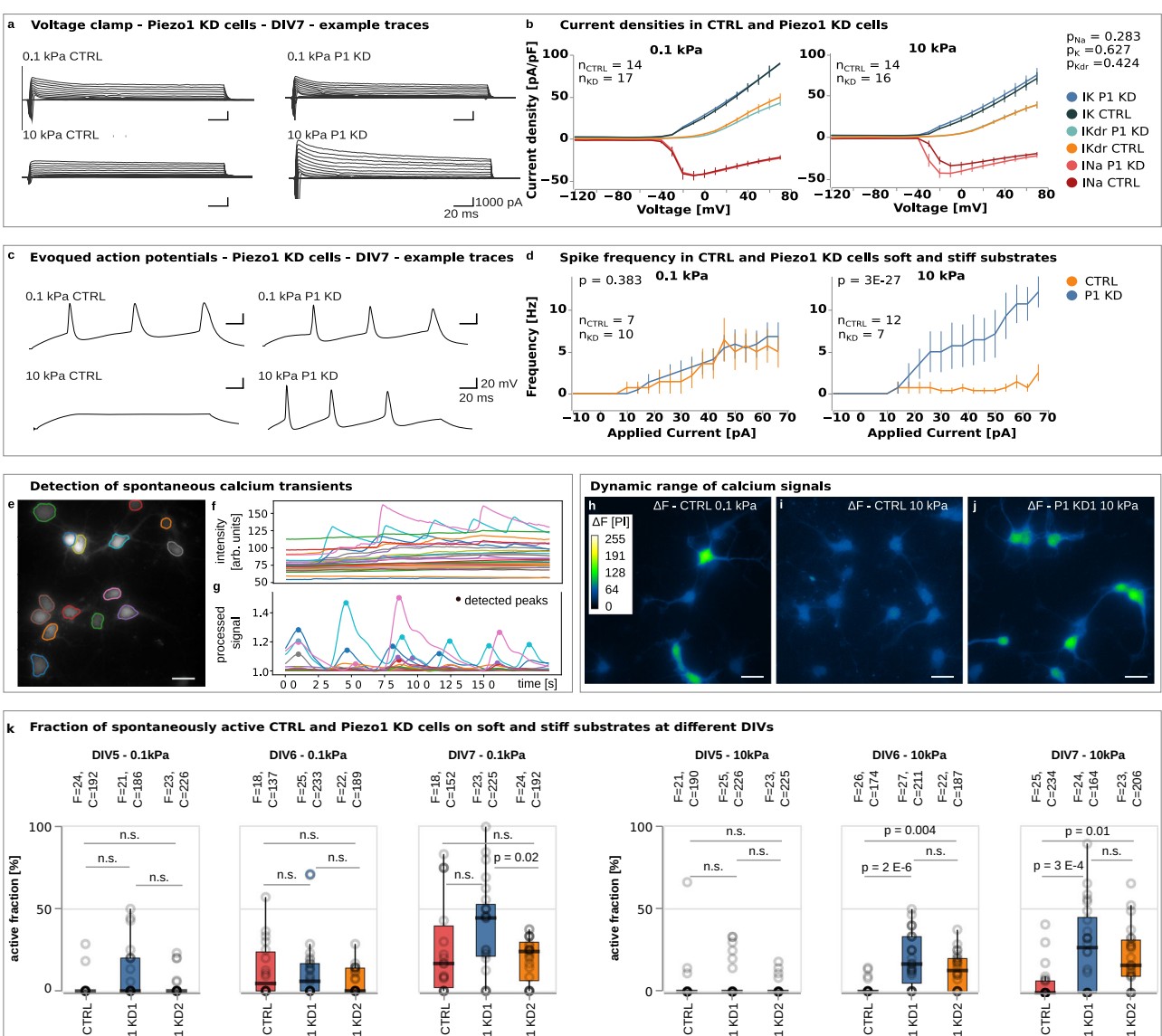

**Fig. 3 | Piezo1 delays the electrical maturation of neurons on stiff substrates.**
**a** Example traces of voltage-clamp recordings showing induced currents in control and Piezo1 knockdown (P1 KD) cells on soft and stiff substrates at DIV 7.
**b** Potassium current densities on soft and stiff substrates and sodium current densities on soft substrates were similar between control and P1 KD cells. On stiff substrates, control cells showed a trend towards lower peak sodium current densities compared to P1 KD cells, suggesting a Piezo1-dependent delay in maturation (see Fig. 2b), though not statistically significant (ANOVA and Sidak's multiple comparisons test). **c** Representative traces of evoked action potentials recorded at DIV 7. **d** Analysis of spike frequencies as a function of injected current. While control and P1 KD cells displayed similar excitability on soft substrates, control neurons exhibited significantly reduced excitability on stiff substrates compared to P1 KD neurons (two-way ANOVA, factors: applied current and genetic condition), further supporting the idea of a Piezo1-dependent delay in neuronal maturation on stiff substrates. *p*-values are indicated in the plots. **e** Image of Rhod-4-loaded

neurons, with semi-automatically labelled somata, and soma fluorescence intensities. **f** Plot of the baseline-corrected intensity of each cell over time. **g** Filtered signals with automatic peak detection. Colour-coded images showing intensity changes ΔF over 40 s. CTRL cells on **h** soft substrates showed strong variability, whereas **i** low fluctuations on stiff substrates indicated little neuronal activity. **j** Knockdown of Piezo1 rescued the spontaneous activity of neurons on stiff gels. **k** Active fraction of cells as a function of substrate and time. Cells were considered active if they produced at least one calcium transient ("F" = fields of view, "C" = number of cells). CTRL neurons (red) on soft substrates became active by DIV 5–6 but showed little activity on stiff substrates until DIV 7. Knockdown of Piezo1 (blue and yellow) rescued the activity of neurons on stiff substrates (Kruskal–Wallis test and two-sided Tukey test), confirming that Piezo1 activity delayed the electrical activity of neurons on stiff substrates. Boxplots show the medians (central line) and interquartile ranges (boxes); whiskers represent 1.5 times the interquartile range. Scale bars: 25 μm. See also Supplementary Movies 1–4.

Fig. 14). These data suggested that an enhanced Piezo1 activity on stiff substrates delayed the electrical maturation of neurons, while Piezo1 knockdown abrogated this delay.

## Piezo1 regulates substrate stiffness-dependent neuronal activity via Transthyretin

To determine potential signalling cascades regulating neuronal maturation downstream of Piezo1, we used RNAseq for an unbiased

comparison of global alterations in CTRL and Piezo1 KD (P1 KD1 and P1 KD2) neurons cultured on soft vs. stiff substrates at DIV 7 (Fig. 4a, b). In CTRL cells, 55 genes were significantly differentially regulated (*p*-value < 0.05, |FoldChange| ≥ 1.1), with 22 genes with differentially increased expression on soft substrates and 33 genes with differentially increased expression on stiff ones.

The RNAseq analysis revealed that the most significantly differentially expressed gene for CTRL neurons cultured on soft and stiff

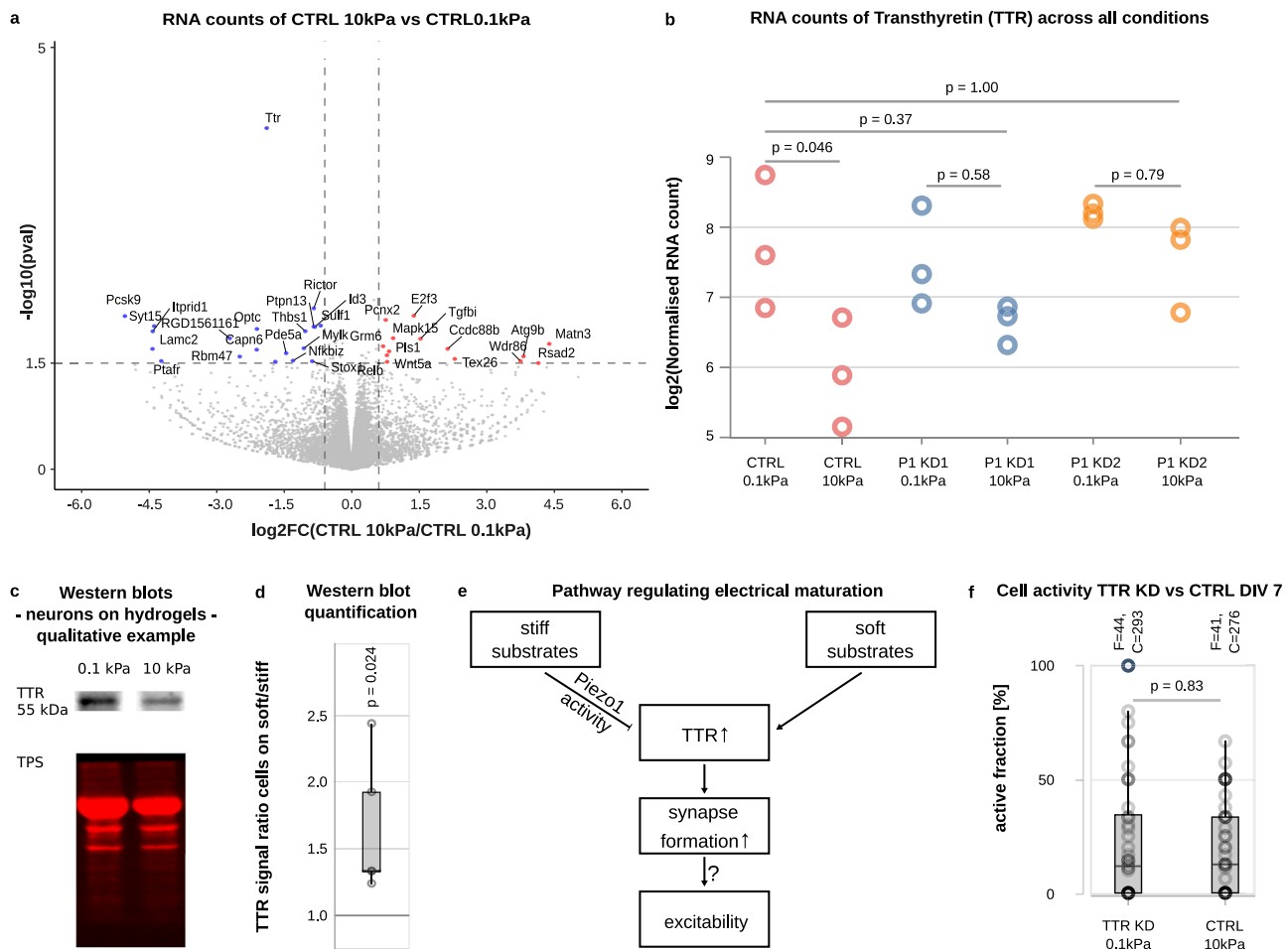

**Fig. 4 | Transthyretin (TTR) regulates electrical maturation of neurons downstream of Piezo1. a** Volcano plot of RNA-sequencing results of CTRL neurons grown on soft *vs.* stiff substrates at DIV 7. Foldchange is shown on the x-axis (blue: higher expression on soft, red: higher expression on stiff gels), *p*-values on the y-axis. Statistical significance was assessed using the DESeq2 two-sided Wald test. The volcano plot displays unadjusted *p*-values. The most significant change between the conditions was observed in *TTR* levels. **b** *TTR* RNA counts normalised by sample-specific size factors for CTRL and Piezo1 knockdown neurons on soft and stiff substrates as assessed by RNA sequencing. *TTR* levels were similarly high in control cells on soft substrates and in Piezo1 KD cells on soft and stiff substrates. However, in control cells cultured on stiff substrates, *TTR* levels were significantly lower, indicating that knockdown of Piezo1 rescued the effect of a stiff substrate. The pattern of *TTR* levels in the different groups resembled that of their activity at DIV 7 (low in CTRL neurons cultured on stiff substrates, high in all other conditions;

Fig. 3k), suggesting a link between *TTR* levels and neuronal maturation (ANOVA test followed by a two-sided Tukey post hoc test). **c** Representative Western blots of CTRL neurons cultured on soft and stiff substrates at DIV 7, stained for TTR and total protein stain (TPS), which was used for normalisation. **d** Analysis of Western blots of CTRL neurons cultured on soft and stiff substrates at DIV 7. The expression of TTR was significantly higher on soft than on stiff substrates ($n = 22$ WB bands from 3 biological replicates; one-sided 1-sample *t*-test). **e** Proposed pathway linking substrate stiffness-dependent Piezo1 activity to the expression of TTR, synapse formation, and potentially intrinsic excitability. **f** Plot of the fraction of active cells in TTR KD neurons on soft substrates and CTRL neurons on stiff substrates assessed by calcium imaging. No significant differences were found, supporting the proposed model shown in (**e**) ($n = 44$ CTRL cells and 41 TTR KD cells; two-sided Mann–Whitney test). Boxplots show medians (central lines) and interquartile ranges (boxes); whiskers represent 1.5 times the interquartile range.

substrates was transthyretin (*TTR*; $p = 6.6 \times 10^{-5}$; 3.7 fold higher expression on soft than on stiff) (Fig. 4a), which we confirmed at the protein level using Western blots (Fig. 4c, d). Moreover, TTR's expression pattern resembled the activity pattern of neurons in the different conditions at DIV 7: We found significantly higher levels of *TTR* and neuronal activity on soft substrates then on stiff ones, and knockdown of Piezo1 led to similarly high *TTR* levels (Fig. 4b) and electrical activity (Fig. 3k) on soft and stiff substrates, indicating a strong relationship between TTR and Piezo1-mediated neuronal maturation.

TTR was originally identified as a thyroxine and retinol transporter[27]. More recently, it has been found to play a regulatory role in the central nervous system by modulating GABA receptor expression[28] and NMDA receptor activation[29]. NMDA receptor activation, on the other hand, increases the expression of glutamate receptors of the AMPA family[30] (cf. Fig. 4e). Together, these findings highlight the central role of TTR in regulating neuronal electrical maturation.

Our results suggested that stiff substrates activate Piezo1, which then suppresses the expression of TTR, leading to delayed neuronal maturation (Fig. 4e). To confirm a role of TTR downstream of Piezo1, we knocked down TTR expression in neurons (Supplementary Fig. 15, "Methods" section) and used calcium imaging to determine the fraction of active cells on soft and stiff substrates at DIV 7. In agreement with an inhibitory role of Piezo1 activity on TTR expression levels, the fraction of active TTR KD cells cultured on soft substrates was as low as that of CTRL neurons cultured on stiff substrates (Fig. 4f), confirming a direct link between substrate stiffness, Piezo1 activity, TTR expression levels and neuronal maturation in vitro.

## Synapse formation in vivo is regulated by tissue stiffness

To test whether tissue stiffness indeed regulates neuronal maturation in vivo, we first mapped the mechanical properties of and synaptic densities in developing *Xenopus laevis* brains at stages 37–38. Brain

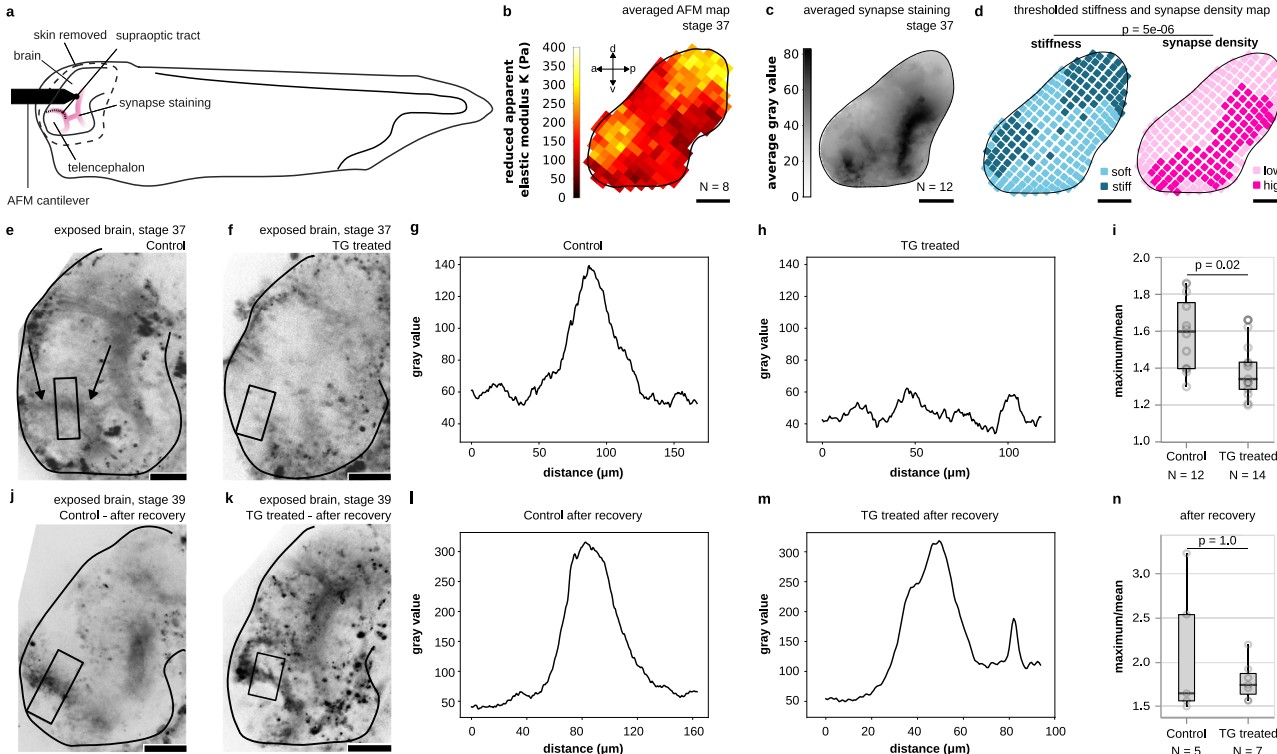

**Fig. 5 | Synapse formation is delayed in stiffer brain tissue in vivo. a** Schematic of a *Xenopus laevis* embryo whose brain was exposed, enabling stiffness measurements by AFM and fluorescence imaging of functional synapses (indicated in pink). **b** Average stiffness map of *Xenopus laevis* brains at stages 37–38. **c** Averaged FM1-43 fluorescence image indicating synapse locations. **d** Binary maps of average tissue stiffness and synaptic density (see "Methods" section). Stiffness negatively correlated with synaptic density ($p = 5 \times 10^{-6}$, two-sided Fisher's exact test). The odds ratio was 0.25, indicating that synapses were four times more likely to form in soft regions than in stiff ones. **e**, **f**, **j**, **k** Images of exposed *Xenopus laevis* brains; synapses were stained using FM1-43. **e** At stage 37–38, a typical staining pattern was found in the area of the supraoptic tract (arrows). **f** In stiffened brains (cf. Supplementary Fig. 15), this pattern was largely absent. **g**, **h** Intensity profiles along the stripes

indicated in (**e**, **f**). While the untreated brain showed a strong peak, the TG profile was noise-dominated. **i** The ratio of the maximum over the mean value in each profile was significantly higher in control brains than in stiffened brains (two-sided Mann–Whitney test), indicating more functional synapses in softer brains. **j**, **k** After a recovery period of 8 h, strong signals in the area of the supraoptic tract were visible in either condition, suggesting that synapse formation was delayed in the stiffened brains. **l**, **m** Line profiles show very similar features between the two conditions at stage 39. **n** The ratio of the maximum value over the mean value showed no significant difference at stage 39 (two-sided Mann–Whitney test). Boxplots show the median (central line), the interquartile range (box), and whiskers represent 1.5 times the interquartile range. Scale bars: 100 µm.

stiffness maps were generated using atomic force microscopy (AFM)-based indentation measurements (Fig. 5a, b), and synapse densities assessed based on fluorescence images of the dye FM1-43, which reports synaptic vesicle recycling in living cells[31] (Fig. 5c). Averaged maps (Fig. 5b, c) were thresholded (Fig. 5d) and compared (see "Methods" section for details). We found a significant negative correlation between the two parameters ($p = 5 \times 10^{-6}$, Fisher's exact test): regions of higher stiffness consistently showed lower synaptic density, confirming our in vitro observations. The odds ratio was 0.25, indicating that in vivo synapses were ~four times more likely to form in soft regions than in stiff regions.

We then tested whether increasing brain stiffness would affect synapse formation in vivo by applying transglutaminase (TG), an enzyme that cross-links extracellular matrix (ECM) proteins. In the developing mouse brain, TG expression is enhanced in the neurogenic niche, which is stiffer than the non-neurogenic brain parenchyma[32], and TG application in heart tissues leads to tissue stiffening[33].

The application of TG to exposed brains of stages 33–34 *Xenopus* embryos for two hours led to a significant stiffening of the tissue in vivo as assessed by atomic force microscopy[14,15] at stages 35–36 (Supplementary Fig. 16). From stages 37–38, brains of wild-type animals showed a characteristic pattern of synapses. This pattern included a semicircle surrounding the telencephalon (Fig. 5a, c, e) coinciding with the position of the supraoptic tract (Supplementary Fig. 17), a structure that is already established by stage 32[34]. In contrast,

in TG-treated, stiffened brains, these patterns were mostly absent at the same stage (Fig. 5f). Quantification of fluorescence intensity line profiles across this region confirmed significant differences between control and stiffened brains (Fig. 5g–i), indicating fewer synapses in this region in stiffened brains. Immunohistochemistry of acetylated tubulin, an axonal marker, revealed that axon tracts formed normally in both conditions (Supplementary Fig. 18), suggesting that the lower number of synapses in the stiffened brains was indeed a consequence of a delay in maturation rather than perturbed axon growth.

To corroborate this hypothesis, stages 37–38 embryos were transferred to *Xenopus* recovery media (Supplementary Methods) and left to develop for another 8 h until stage 39. Post recovery, both groups displayed similar patterns of FM1-43 fluorescence (Fig. 5j, k) and peaks in their intensity profiles (Fig. 5l–n), indicating that synapses had now formed, and thus confirming that the stiffening of the brains indeed delayed synapse formation in vivo.

## Discussion

Here, we have shown that the electrical maturation of neurons is regulated by environmental stiffness. The electrical activity and synapses of primary neurons developed faster in softer environments than in stiffer ones. Knockdown of the mechanosensitive ion channel Piezo1 abolished the delay in neuronal maturation on stiff substrates. We identified Transthyretin (TTR) as a key player in regulating neuronal maturation downstream of Piezo1. On soft substrates, TTR

expression levels were higher than on stiff substrates, which likely promoted synapse formation. Knockdown of Piezo1 resulted in an increase in TTR levels on stiff substrates, which was accompanied by an increase in electrical neuronal activity. As changes in synapse densities can trigger homeostatic adjustments in intrinsic excitability[35], we hypothesise that this way TTR may indirectly link substrate mechanics to the maturation of the intrinsic excitability of neurons (Fig. 4e).

While previous studies have identified environmental stiffness as an important regulator of axon growth and guidance[14,15] as well as cell morphology[16,17,21], our findings demonstrate that stiffness also regulates the timing of the electrical maturation of neurons. Furthermore, we identified TTR as a downstream mediator of Piezo1, thus providing a mechanotransduction pathway linking mechanical cues to the maturation of neuronal excitability.

Neuronal growth cones exert higher traction forces and tension on their axon on stiffer substrates than on softer ones[21,36,37]. In fibroblasts, Piezo1 activity scales with traction forces[25], suggesting that the activity of Piezo1 is also higher in axons grown on stiffer substrates. Our data indicate that this increased activity of the mechanosensitive ion channel on stiff substrates leads to a downregulation of TTR, thus slowing down neuronal maturation.

How Piezo1 activity leads to a decrease in TTR remains to be studied. As a non-selective cation channel, Piezo1 activation gives rise to calcium transients. Piezo1-mediated calcium influxes on stiffer substrates could activate calcium-dependent signalling cascades, potentially leading, for example, to the phosphorylation of repressive transcription factors and subsequent downregulation of TTR expression. Alternatively, Piezo1 activity could lead to an activation of the PI3K/Akt pathway[38], which antagonises TTR activity by inhibition of GSK3[39]. This pathway could lead to a downregulation of synaptic proteins, which could explain the delay in synapse formation on stiff substrates (Fig. 1) and in stiffened *Xenopus laevis* brain tissue (Fig. 5).

Stiffening of brain tissue by transglutaminase delayed synapse formation (Fig. 5e–n). Although perineuronal nets – structures critical for stabilising mature synapses – have not been observed in *Xenopus* embryos at the early developmental stages investigated here, we cannot fully exclude that transglutaminase treatment may also contribute to limiting synaptogenesis via effects on perineuronal nets. However, together with our data showing a strong negative correlation between local brain stiffness and synapse densities in vivo (Fig. 5a–d) and a significant Piezo1-TTR-mediated delay in the maturation of neurons cultured on stiff substrates (Figs. 1–4), these data strongly indicate that the timing and spatial patterning of the electrical maturation of neurons are at least in part regulated by the mechanical properties of the brain. Brain mechanics might thus contribute to region-specific maturation rates and circuit formation in the brain, which has important implications for development and disease, as brain tissue is spatially heterogeneous in stiffness[14] (Fig. 5b), and local tissue stiffness changes during development[15,40–42] and in neurodevelopmental disorders[43].

As development progresses, brain tissue stiffness increases[41,44]. This increase in stiffness may act as a stop signal for neuronal maturation, reducing high rates of synaptogenesis. Such a constraint on synapse formation might be critical for ensuring the specificity and precision of connections required for proper neural circuit function. Our study indicates that local brain tissue stiffness regulates the onset and rate of electrical maturation in each brain area and at different developmental time points, thereby potentially establishing the foundation for the emergence of the first circuits in the developing brain.

Brain tissue stiffness is determined by multiple factors, including the stiffness, density and connectivity of the cells found in the area and the composition of the ECM[13,45]. Alterations in any of these components may thus impact neuronal maturation during brain development. Similarly, impaired mechanosensitivity, for instance, through

altered expression levels or mutations of mechanosensitive proteins, might affect the maturation speed of neurons and consequently alter the emerging circuitry. Knowledge about when and where the regulation of neuronal maturation by tissue stiffness is critical during brain development will help understand the emergence of healthy neural circuitry and potentially also the onset of different neurodevelopmental disorders, and it may contribute to a better understanding of the decline in neuronal function in the ageing brain, whose mechanical properties change with age and in neurodegenerative diseases[13].

## Methods

All experiments involving Sprague Dawley rats and *Xenopus laevis* were conducted in compliance with the Ethical Review Committee of the University of Cambridge and the UK Home Office. Xenopus laevis embryos were used prior to stage 45, the onset of sexual differentiation. Brain tissue from rat embryos was collected at embryonic days 17–18 (E17-18), and tissue of all embryos from the same litter was processed together. Consequently, rat neurons were of mixed sex origin.

A complete list of key resources is available in Supplementary Data 2.

### Statistics & reproducibility

At least 3 biological replicates were used for every experiment. Randomisation was not applicable, as groups were defined by treatment or genotype and processed in parallel. The order of data collection was randomised. The Investigators were not blinded to allocation during experiments and outcome assessment as experimental conditions (e.g., genotype or treatment) were apparent during data collection. However, all data were analysed automatically, thus avoiding any potential bias. Sample sizes were chosen based on experience; no statistical method was used to predetermine sample size. Data were only excluded if an animal or a sample died during the measurement. Unless stated otherwise, normally distributed data were analysed using two-sided t-tests or ANOVAs for comparing two groups or more than two groups, respectively. For non-normally distributed data, Mann–Whitney and Kruskal–Wallis tests were used for comparing two groups or more than two groups, respectively. Boxplots show the median (central line), the interquartile ranges (boxes), and whiskers represent 1.5 times the interquartile range.

### Preparation and coating of hydrogels

Hydroxy-polyacrylamide substrates were fabricated as described in ref. 14. Glass-bottom Petri dishes, as well as glass coverslips with 18 mm diameter, were washed in 70% ethanol, followed by distilled water and then air dried. The glass bottoms (18 mm coverslips for synapse stainings as well as patch-clamp experiments; glass-bottom Petri dishes for all remaining experiments) were wiped with 1 M NaOH, allowed to dry and incubated for 3 min with (3-aminopropyl)trimethoxysilane (APTMS, Sigma Aldrich, 281778). APTMS was then washed off, and the glass bottoms were incubated with 0.5% glutaraldehyde for 30 min. The glass bottoms were then washed three times with distilled water. The coverslips were incubated with Rain-X (Shell Car Care International) for 10 min. To make the acrylamide (AA, Sigma Aldrich, A4058) stock solution, 500 µl of 40% acrylamide was mixed with 65 µL of 100% hydroxyacrylamide (Sigma Aldrich, 697931). To prepare the gel premix, 500 µl of acrylamide stock solution was mixed with 250 µl of 2% bis-acrylamide (BA, Fisher Scientific, BP1404-250). For 0.1 kPa hydrogels, 53 µl of the premix was diluted with 447 µl PBS, and for 10 kPa hydrogels, 150 µl was diluted with 350 µl PBS. These mixtures were then placed in a vacuum chamber for 4–6 min. Then, 1.5 µL of 0.3% (v/v) N,N,N′,N′-tetramethylethylenediamine (TEMED, Thermo Fisher, 15524–010) and 5 µL of 0.1% (w/v) ammonium persulfate (APS, Sigma, 215589) solution were added to the gels and mixed. Next, 14 µL

of the gel mixture was pipetted onto each glass bottom and covered with a Rain-X-treated coverslip. The dishes were then filled with 2 mL PBS and left for 20 min before the coverslips were removed. The gels were washed twice with sterile PBS in a tissue culture hood, treated with UV light for 30 min and washed again with PBS. Subsequently, the gels were coated with 10–100 µg/ml PDL in PBS or Borate buffer (for patch-clamp experiments) overnight at 4 °C, washed three times with PBS, and finally coated with 1–2 µg/ml laminin in Neurobasal media (NB media, Gibco) with 1% PSF at 37 °C for 4 h.

## Tissue dissection

Pregnant Charles River Sprague Dawley rats were sacrificed at E17 or E18. They were anaesthetised with 5% isoflurane before cervical dislocation and exsanguination. The uteri were removed and transferred into a sterile 100 mm culture dish. The dish was transferred to a sterile horizontal flow hood. The embryonic sacs were opened in a new dish, and the embryos were transferred into a separate sterile dish. The embryos were decapitated with a razor blade and the heads transferred to a fresh dish with chilled HBSS+ (HBSS w/o Ca, w/o Mg (Gibco) with 2% penicillin-streptomycin-amphotericin B (PSF; Lonza)) and kept on an ice pack. The skulls were opened from the neck to the forehead, and the brains removed with curved forceps. The brains were moved into a dish containing chilled Hibernate E+ (Hibernate E (Gibco) with 2% PSF) and kept on an ice pack. Meninges were removed, and the hippocampi were cut out. The hippocampi were collected in 2 ml Eppendorf tubes containing Hibernate E+ and kept in a cooling rack until the dissection was completed.

## Neuronal culture

**For Patch Clamp and synapse staining experiments of wild-type neurons**, hippocampal tissue was washed in HBSS with 2% PSF, and then incubated with Papain solution for 30 min at 37 °C. The solution was removed, and 1 ml Ovomucoid was added to stop the papain digestion (Supplementary Methods). The step was repeated before gentle trituration with a 1 mL pipette. The cell suspension was topped up to 8 ml with Ovomucoid before centrifugation at 1000 RPM for 8 min. The supernatant was removed, and the cells were suspended in 1 ml neuron culture medium (Supplementary Methods). After 24 h, half the volume of the media was replaced. Afterwards, half the volume of the media was replaced every 2/3 days.

**For wild-type (WT) cell culture**, hippocampal tissue was washed 3× with cold HBSS+. The buffer was removed and replaced with 2 ml 0.05% prewarmed Trypsin-EDTA (Gibco) and incubated for 10 min at 37 °C. The tissue was washed 10× with warm HBSS+.

The HBBS+ was then replaced with 1 ml warm Neurobasal complete media (Supplementary Methods). Cells were isolated by gently pipetting up and down until most tissue bits were broken down. 150,000 cells were plated per dish in 1.5 ml Neurobasal complete. After 6–12 h, the media was replaced completely. Afterwards, half the volume of the media was replaced every 3 days.

For CRISPR knockdown (KD) and control cells, the CRISPR reagents were prepared before washing the tissue.

**For the Piezo1 KD cells**, two different sets of guides were prepared. Each set contained 4 different guides that are specific to different regions of the gene (see below). As a control, TRACR RNA was used.

120 pmol of each single-guide RNA (IDT) was incubated with 52 pmol HiFi Cas9 (IDT) in 2.1 µl nuclease-free buffer (IDT) and incubated for 10–20 min at room temperature. Per condition, the 4 specific guide mixes were transferred into 100 µl of electroporation buffer (82 µl electroporation buffer + 18 µl supplement from the Lonza rat nucleofector kit). Additionally, 400 pmol electroporation enhancer (IDT) was added.

**For the control cells**, 240 pmol of the TRACR RNA (IDT) were incubated with 208 pmol HiFi Cas9 (IDT) in 4.2 µl nuclease-free buffer

(IDT) and incubated for 10–20 min. This mixture was transferred into 100 µl of electroporation buffer, and 400 pmol electroporation enhancer (IDT) was added.

**For the TTR KD cells**, two guides were designed (the gene is very short and thus very few target regions were available in the gene). 120 pmol of each single-guide RNA (IDT) was incubated with 104 pmol HiFi Cas9 (IDT) in 2.1 µl nuclease-free buffer (IDT) and incubated for 10–20 min, and then transferred into 100 µl of electroporation buffer, and 400 pmol electroporation enhancer (IDT) was added.

For each condition, hippocampal tissue was washed 3× with cold HBSS+. The buffer was removed and replaced with 2 ml 0.05% prewarmed Trypsin-EDTA (Gibco) and incubated for 10 min at 37 °C. The tissue was washed 10× with warm HBSS+. Then, the tissue was transferred into the electroporation mix, and the cells were isolated by gentle pipetting. The cell suspension was transferred into the electroporation cuvette (Lonza rat nucleofector kit) while avoiding bubbles. The cuvette was placed in the nucleofector 2b (Lonza) and the programme O-003 was executed. The cuvette was immediately topped up with 1 ml prewarmed Neurobasal complete media. The content of the cuvette was transferred into a 1.5 ml Eppendorf tube, and the cells were counted and plated as described above.

## CRISPR guide design and characterisation

**Piezo1 KD guides**. Two distinct CRISPR-CAS9 knockdown (KD) conditions were established. Each condition was based on four different guides. The eight guides were distributed into sets 1 and 2 in alternating order along the gene (set 1: #1, #3, #5, #7; set 2: #2, #4, #6, #8, see below).

Each guide was designed to target a specific region of the *Piezo1* gene. Following transfection with the CRISPR-Cas9-guide complex, the complex binds to its complementary DNA sequence. Cas9 then introduces a double-strand break near the protospacer adjacent motif (PAM) site. During the repair process, errors occur that can result in base pair changes or frameshift mutations. Not all guides induce genetic changes in every cell. Since multiple guides were used simultaneously per knockdown condition, each acting independently, this approach resulted in a heterogeneous cell population.

To estimate the editing efficiency of the CRISPR-Cas9 guides used, one guide per set was selected (guide 7 for set 1 and guide 8 for set 2), and the DNA of the experimental and control conditions around the guide of interest were sequenced.

DNA from the control and experimental conditions was extracted and amplified using TOPO cloning. Finally, several bacterial colonies were picked, each containing only a single sequence. The DNA from each colony was extracted and sequenced using Sanger sequencing.

For set 1, DNA from 8 colonies was sequenced, containing DNA from the control condition and from 15 colonies from the experimental condition. 5 of the control sequences were mutation-free, 3 sequences could not be used as they only contained the sequence of the cloning vector. Out of 15 sequences obtained from KD set 1, we found 4× point mutation, 9× mutation-free, and 2 sequences could not be used as they only contained the sequence of the cloning vector (31% editing).

For set 2, DNA from 4 colonies was sequenced, containing DNA from the control condition and from 10 colonies from the experimental condition. The control sequences were all mutation-free. Out of 10 sequences obtained from KD set 2, we found 1× point mutation, 1× 7 base mutation, 1× deletion, 7× mutation-free (30% editing).

Assuming that all 8 guides have similar editing efficiencies ($P_{mut} = 0.3$), and in each set, 4 guides act independently, the probability of not finding any mutations corresponds to $P_{WT} = (0.7)^4 = 0.24 = 24\%$ (see Supplementary Fig. 10). This suggests a probability of finding at least one mutation in the *Piezo1* gene with a probability of $P_{mut} = 1 - 0.24 = 0.76 = 76\%$.

Given that no two mutations we found in the sequences were the same, we assume an accumulation of random point mutations and

occasional frameshifts in the *Piezo1* gene. While the frameshifts lead to lower expression levels of Piezo1, the point mutations lead to changes in amino acids, which might change the functionality of the mechanosensitive protein, e.g., through changes in folding.

Guide sequences were as follows. Set 1 used odd-numbered guides, set 2 used even-numbered guides:

#1 GTTGAGACACGTTTGCCCAA (PAM: AGG, exon: 4)
#2 ACGCTTCAATGCTCTCTCGT (PAM: TGG, exon: 2)
#3 TCCTTGTGAGGCGTCCACAG (PAM: AGG, exon: 5)
#4 CCACCCTGGCAACCAAACGC (PAM: AGG, exon: 6)
#5 CTTTAACACCCTCTGCGTCA (PAM: AGG, exon: 7)
#6 GAACACCATCAGGTAGACGC (PAM: TGG, exon: 7)
#7 GTACTCCAGTAACTGCACCG (PAM: AGG, exon: 19)
#8 CGATTTTGTAGACCACCAGG (PAM: CGG, exon: 14)

**TTR guides.** For the TTR KD, one CRISPR-CAS9 KD condition was established. Given the very short gene, two guides were designed. To estimate their editing efficiency, DNA was sequenced in the experimental and control conditions around both sequences of interest as described for the Piezo1 KD cells.

For guide 1, DNA from 9 colonies was sequenced, containing DNA from the control condition and from 35 colonies from the experimental condition. Out of the 9 control sequences, 8 were mutation-free, and 1 sequence could not be used as it only contained the sequence of the cloning vector. Out of 35 sequences obtained from the experimental condition, 34 were mutation-free, and 1 sequence could not be used as it only contained the sequence of the cloning vector. (0% editing).

For guide 2, DNA from 8 colonies was sequenced, containing DNA from the control condition and from 12 colonies from the experimental condition. Out of the 8 control sequences, 6 were mutation-free and 2 sequences could not be used as they only contained the sequence of the cloning vector. Out of 12 sequences obtained from the experimental condition, 7 were mutation-free, 1 had a point mutation, 1 sequence had an insertion, and 3 sequences could not be used as they only contained the sequence of the cloning vector. (22% editing). KD samples show a lower TTR expression than the CTRL (see Supplementary Fig. 15).

The guide sequences were:
TTR guide #1: AAGAGCCTTCCAGTACGATT (PAM: TGG)
TTR guide #2: GTTTTCACAGCCAATGACTC (PAM: TGG)

**Immunocytochemistry of synapses in neuronal cultures and imaging**

The samples were fixed with 4% PFA for 10 min before being washed with PBS. To block and permeabilize the samples, they were incubated for 1 h in 0.1% Triton X-100 and 10% goat serum in PBS at 21 °C. Subsequently, the samples were incubated with the primary antibody overnight at 4 °C, followed by incubation with the secondary antibody for 1 h at 21 °C. The primary antibodies used were chicken anti-Neurofilament (abcam, ab4680, 1:1000), rabbit anti-neuroligin (Synaptic Systems #129 213, 1:1000), guinea pig Vglut2 (Synaptic Systems #135 404 1:1000), and mouse VGAT (Synaptic Systems #131 011, 1:750). The secondary antibodies were goat anti-chicken IgY H&L (Alexa Fluor 405, abcam, ab175674, 1:1000), goat anti-rabbit IgG-488 (Life Technologies, A-11034), goat anti-guinea pig IgG-568 (Life Technologies # A-21435), and mouse IgG-647 (Life technology, catalogue number unknown, 1:1000). Images were taken on a confocal microscope (Leica, SP8) with a 63x objective (NA = 1.4).

**Synapse density analysis of immunocytochemistry images**

To analyse synapse density, a custom image analysis code was used. First, a maximum intensity projection was performed for all four channels. Subsequently, spot detection was carried out on the neuroligin, vesicular GABA transporter, and the vesicular glutamate transporter channels. This was achieved by enhancing the bright, circular features using an à trous wavelet transform, which is based on order three B-splines and has three detail levels. As the images consist of a dark background superimposed with bright spots, the majority of the computed wavelet coefficients represent the background and were modelled at each wavelet scale as samples from Gaussian distributions with a mean of 0. The signal pixels appeared as outliers in these wavelet coefficient distributions. The outliers were identified by controlling the false discovery rate at a 1% level[46] across the detail levels. The pixel positions classified as outliers across all scales constitute the support of the signal (the spot representing the molecules). To distinguish between close molecules with overlapping support, the local maxima of the original image were counted within the computed support (see Supplementary Fig. 1). This approach was inspired by ref. 47.

The Frangi vesselness filter[48] was applied to enhance the images of processes. Segmented images of processes were obtained by applying a similar support computation to the spot detection, based on the wavelet transform. Cell bodies were detected in the same channel using Otsu thresholding and separated from processes through morphological opening. The length of a process in each image was measured by summing all pixels in the resulting process mask after removing the cell body-related area.

If the distance between a neuronal process (NF filament), GABA or glutamate transporter signals and neuroligin was larger than 2 pixels (360 nm), the signals were excluded. Counts were normalised to the total process length and presented in boxplots.

**Sholl analysis**

Neurons were cultured on soft and stiff substrates for 24 h. Half of the media was then replaced with prewarmed (37 °C) 4% PFA and 7.5% sucrose solution and incubated for 30 min at room temperature. Samples were washed nine times with PBS, followed by permeabilisation for 5 min with 0.1% Triton X-100. Cells were stained with DAPI (Merck, D5942; working concentration: 0.1 μl/ml) in PBS containing 0.0001% Triton X-100 for 20 min. Finally, samples were washed six times with PBS and imaged on a Leica DMI8 microscope using a 63× oil-immersion objective (NA 1.4).

For image analysis, LIF files were converted to 8-bit TIFF images using the Python readlif library. Nuclei were segmented from the fluorescent channel using Otsu thresholding (Python skimage library) and converted to binary masks. Objects smaller than 1000 pixels² were excluded. Brightfield images were processed to generate binary masks of the cells using a Meijering neuriteness filter (skimage) with sigma values of 2, 3, and 4, followed by thresholding with the triangle algorithm (skimage) and removal of speckles smaller than 50 pixels². For Sholl analysis, the centre of each nucleus was used as the origin, and the distance from each skeleton pixel to this centre was calculated. For each Sholl radius (starting at 20 μm with 10 μm intervals), all skeleton pixels within ±1 pixel of the target radius were selected. These pixels were then dilated using a disk of radius two pixels to close small gaps within the neurites. Intersections between the Sholl radii and the skeletonised cell masks were determined and marked in light blue (Supplementary Fig. 2a–d). Finally, malformed cells (i.e., cells dead prior to fixation), cells contacting the edge of the field of view, and cells overlapping with neighbouring cells were manually excluded.

**Patch-clamp recordings**

Whole-cell patch-clamp recordings were performed using a Multiclamp 700B amplifier (Molecular Devices) connected to a 16-bit digitiser (Axon Digidata 1440A, Molecular Devices), using a CV-7B headstage (Axon Instruments). The software for data acquisition was pClamp 10 with a sampling rate of 50 kHz and a 10 kHz low-pass filter. Cells were continuously perfused with oxygenated artificial cerebrospinal fluid (ACSF) consisting of 144 mM NaCl, 2.5 mM KCl, 2.5 mM

CaCl$_2$, 1 mM NaH$_2$PO$_4$, 10 mM HEPES, and 10 mM glucose, with pH adjusted to 7.35 using NaOH. The electrodes were filled with an internal recording solution containing 130 mM K-gluconate, 4 mM NaCl, 0.5 mM CaCl$_2$, 10 mM HEPES, 10 mM BAPTA, 4 mM MgATP, 0.5 mM Na$_2$GTP, and 2 mM K-Lucifer yellow, with pH adjusted to 7.3 with KOH and an osmolarity of 295 mOsm. The series resistance ranged from 5 to 20 MΩ, and the electrode junction potentials (-14 mV) were corrected for.

Upon finding a cell, a single -5mV voltage pulse was applied for 50 ms to generate cell membrane capacitive transients, so that series resistance, membrane resistance and cell membrane capacitance could be calculated. The time constant of a single exponential curve was fitted to the transients, and the cell membrane properties were calculated using MATLAB. A voltage step protocol was used to examine ion channel currents. A series of 20 200 ms, 10 mV voltage steps, from −120 mV to +70 mV, was applied to the cell, and the sodium and potassium current responses were recorded. Between the steps, the cells were held at a baseline voltage of −60 mV. The peak sodium current ($I_{Na}$), the potassium current ($I_K$) at +40 mV, and the delayed rectifier potassium current ($I_{Kdr}$) at +40 mV were detected using a custom Python script. The current densities of the voltage-gated ion channel currents were calculated by dividing the resultant current by the cell membrane capacitance. Group differences were assessed via Two-Way ANOVA followed by two-sided Sidak's multiple comparisons test, using Python.

Spontaneous and evoked action potentials were recorded in current-clamp recording configuration with a 1 kHz sampling frequency. To record spontaneous action potentials, baseline current was injected to hold the membrane potential at -60mV, gap-free recordings were carried out for up to 10 min and stopped when the first APs were recorded. As an additional quality control step, the Lucifer yellow signal was inspected after each recording to check for potential leakage. A cell was considered active if it produced at least one action potential during the recording. To record evoked action potentials, a series of 200 ms current steps was applied, and the resulting current response was recorded. For WT cells at DIV 5 and DIV 7, current steps ranging from −10 to 30 pA were injected (Fig. 2k). At DIV 10 and DIV 14, the range was extended to −60 to 140 pA to reliably evoke action potentials (Fig. 2k). For P1 KD and control cells at DIV 7, the range was adjusted to −10 to 70 pA, to ensure reliable AP generation (Fig. 3d).

If no events were visible in the first attempt, the step size was increased from 2 pA to 10 pA. Both spontaneous and evoked action potentials were detected and quantified using a MATLAB or Python script. Action potentials were identified based on three criteria: (i) the signal had to exceed a minimum amplitude threshold, (ii) overshoot 0 mV to indicate full reversal of membrane potential, and (iii) be followed by a clear afterhyperpolarization. In our analysis script, these criteria were implemented using a minimum peak prominence of 20 mV, a minimum peak height of 0 mV, a maximum peak width of 60 ms, and a minimum inter-peak interval of 20 ms. All automatically detected peaks were manually reviewed to ensure they met these criteria.

## Calcium imaging

50 μg Rhod-4™ (AAT Bioquest) were dissolved in 49.21 μl DMSO to generate a 1 mM stock and aliquots were kept at −20 °C.

Neurons were incubated with 4 μM Rhod-4 (3 μl stock solution in 750 μl Neurobasal complete media, see Supplementary Methods) for 30 min at 37 °C, 5% CO$_2$, 100% humidity. Cells were washed three times with prewarmed media. And left to recover for 45 min at 37 °C, 5% CO$_2$, 100% humidity.

Each dish was imaged at 37 °C for less than 30 min with a Leica DMi8 using a ×63 oil objective (numerical aperture 1.4) and a YFP filter set. Approximately 10 fields of view were imaged per dish while avoiding cell clusters. For each FOV, we recorded one picture every

80 ms for approximately 40 s. Files were saved in .lif format. Cell health was assessed by visual inspection. Cells that showed morphological alterations indicating pathological changes, such as blebbing, beading, or poor cell-to-substrate adherence, were not used for experiments.

## Tetrodotoxin treatment

Tetrodotoxin (TTX) treatment was performed to test calcium peaks for dependence on voltage-gated sodium channels at DIV 7. For TTX treatment, cells were exposed to 0.1 μM TTX for 2 min prior to imaging. In contrast to the control cells, none of the TTX-treated cells showed any activity (see Supplementary Fig. 13). This suggests that the calcium peaks we detected were triggered by action potentials.

## Analysis of calcium imaging data

lif files were converted into .tiff & .png files with a custom Python script utilising the readlif library. Cell labelling initially used supervise.ly and later Label Studio. A neural net based on the U2-Net[49] was trained on manually labelled images and used for automated segmentation. The automated segmentation results were manually validated and adjusted (Fig. 3e).

Cell activity was quantified using our custom Python software, which is available at https://gitlab.com/rknt/cell-activity-from-calcium-imaging. Cell segmentation data was read, as well as the .tiff files from each FOV. The average fluorescence intensity was determined within each cell outline as a function of time (Fig. 3f). For each intensity profile, a polynomial baseline was fitted (using peakutils.baseline), which correlates with the bleaching of the fluorescent dye inside the respective cell. The fluorescent intensity was normalised using the baseline and smoothed using a Butterworth filter (using scipy.signal.butter) (Fig. 3g). Then a peak finder (scipy.signal.find_peaks) was applied to the filtered data, and the peak prominence, the peak width and peak length of each peak were determined. In addition, the variance of each normalised trace was determined.

Cells were classified as "active" if they had at least one peak that exceeded certain threshold criteria (Fig. 3g) and variance in their normalised signal that exceeded a threshold variance (see Supplementary Fig. 11). The thresholds were optimised using a training set of 77 calcium imaging series that had been manually analysed (see Supplementary Fig. 12). Using the following values automated analysis reached a sensitive of 94% and a specificity of 99% when compared to the manual analysis: peak prominence = 0.023, peak width = 0.1, maximum peak length = 37, and variance = $10^{-4}$. These values were used for all subsequent quantification of cell activity. The active fractions of FOVs across conditions were compared using a Kruskal–Wallis test and Tukey post hoc test.

In order to assess the efficacy of the automated analysis procedure, a manual analysis was conducted on 92 FOVs comprising 1281 cells. This involved viewing the videos and noting in a spreadsheet whether or not each cell exhibited observable activity. The manual and automatic results were then compared in terms of the number of active cells in each FOV. During this process, it became evident that our manual approach was prone to errors. Upon conducting a comparison between manual and automated analyses, we discovered numerous active cells that had escaped manual detection. Upon reviewing the videos, we realised that the discrepancies largely stemmed from instances where we had visually overlooked certain activities. Consequently, we corrected the spreadsheet where necessary and repeated the process. Supplementary Fig. 12 shows the difference between the numbers of active cells that we determined to be active by visual inspection and automated analysis. In 76 out of 92 FOV, the two methods agreed on the activity in every single cell. Analysing the sensitivity *sn* and specificity *sp* of the signal detection (assuming the manual analysis as ground truth), we found *sn* = 96%, and *sp* = 98%.

## Western blots

**Experimental procedure WBs.** All buffers and media used in this protocol are listed in the Supplementary Methods. WT hippocampal neurons were grown on hydrogels as described above. At DIV 7, one dish at a time was processed: cells were washed once with PBS (at RT), covered with 300 μl ice-cold Magic RIPA Buffer media containing 1% protease and phosphatase inhibitor cocktail and left on ice on a shaker for 20 min. After 20 min, the next dish of identical background was washed with PBS, and the lysate buffer from the first dish was carefully pipetted to the second dish, which was then left on the shaker on ice for 20 min. This procedure was repeated with all 15 dishes in the same condition. In the end, the lysate buffer was transferred into a 1.5 ml Eppendorf tube and kept at −80 °C for further processing.

The protein concentration was measured using a Bradford Assay. The samples were diluted to 1 mg/ml and 0.5 mg/ml protein concentration in LDS sample buffer (Nupage NP007) with 5% (v/v) beta-mercaptoethanol, respectively (see below). The samples were denatured on a heat block at 95 °C for 5 min before loading them onto the PAGE gels.

Depending on the size of the protein, either a 4–20% or a 4–12% gradient gels were used for the blots. Each lane was loaded with 20 μL sample or with the protein ladder. Gels were run at RT at 40 mA using MOPS SDS running buffer.

The gels were washed in transfer buffer and blotted using the Biorad Mini PROTEAN Transfer System in a cold room (4 °C) for 16 h at 21 V.

For probing, the membranes were dried for 1 h at RT, and all following procedures were performed on a shaker at RT. After wetting with TBS for 5 min, the membranes were blocked in 5% milk in TBS for 30 min and probed with the appropriate antibody dilution in 5% milk in TBST for 1 h. After 3 washes with TBST, the membranes were probed with the appropriate HRP-conjugated secondary Antibody for 1 h in 5% milk in TBST, and washed in TBST before the ECL reactions were carried out. Imaging was performed using the Li-Cor Odyssey FC. For Total Protein Stains (TPS), the Li-Cor Revert 700 reaction was performed according to user instructions, and the membranes were imaged using the 700 nm channel of the Li-Cor Odyssey FC.

Antibodies used in these WBs at the following concentrations:
Piezo1, rabbit, polyclonal (Novus Biologicals, NBP1-78446, 1:1000)
TTR, rabbit, polyclonal (Invitrogen, PA5-27220, 1:500)
Goat anti-rabbit (abcam, ab97080, 1:10000)

## WB analysis

TTR and Piezo1 bands were normalised using total protein staining. The protein bands were analysed using Image Studio Lite by Li-Cor. The resulting data were saved in .csv files and normalised, and plotted with a custom-written Python script. To test the signal quality in our Western blots, we compared the signals of the lanes loaded with lysates of 1 mg/ml protein concentration (normal concentration) and those loaded with lysates of 0.5 mg/ml protein concentration (low concentration). If the signals were in the linear range, the normalised signals of the normal protein concentration lanes would be expected to be similar to those of the low concentration lysates.

## Analysis of WBs for pathway analysis

First, WB bands were analysed as described above. Then, the ratio of the WB signals between lysates of cells grown on soft gels and lysates from cells grown on stiff gels was calculated. Subsequently, the ratio $r$ was evaluated:

$$r = \frac{normalised\,signal_{soft}}{normalised\,signal_{stiff}} \tag{1}$$

Multiple biological replicates were processed ($n > 3$), each with multiple blots and technical replicates. To calculate **r**, the median of

the normalised signal for each band was determined for each biological replicate, blot, technical replicate, and lysate concentration. The ratio was then built between the median from the lysate of cells grown on soft gels and the lysates from cells grown on stiff gels.

If **r** < **1** for the lysate of one protein concentration but **r** > **1** for the lysate of the other protein concentration on the same blot, the blot was discarded.

## Specific analysis of the Transthyretin bands

Transthyretin (TTR) was blotted on two different gels: 4–20% gels and 4–12% gels.

In its functional form, TTR is a tetramer (dimer of dimers). The fraction of these tetramers which was broken down to its monomeric form during the linearisation process in the WB protocol is unclear. We expected to see 3 bands associated with the monomer, the dimer and the tetramer. In order to blot all these bands, we found that we had to combine two different gel substrates, 4–20% gels for the low molecular weight bands (usually only mono and dimer) and 4–12% gels for the higher molecular weight bands (usually only dimer and tetramer).

Since the strength of the signals corresponding to the monomer and dimer was very weak compared to the tetramer and could often not be quantified, we concluded that most tetramers were not degraded. Therefore, only tetramer bands were considered in the pathway analysis experiments.

## RNA purification

RNA was purified using the Qiagen RNeasy Plus Micro Kit (#74034).

Hippocampal neurons were cultured (as described above) on hydrogels of different stiffnesses for 7 days. The culture medium was removed, and 350 μl RLT plus buffer (Qiagen Kit #74034) containing beta-mercaptoethanol was added. The gels were scraped from the dish and transferred to a 1.5 ml Eppendorf tube. Each sample was vortexed for 30 s and then homogenised by pipetting up and down (x50) using a cut pipette tip. Afterwards, the RNA purification protocol was carried out as described in the manual and total RNA was diluted with 20 μl of RNase-free water. The 70% and 80% ethanol, as well as the RLT plus beta-mercapto buffer, were freshly prepared. The RNA concentration was determined using a nanodrop. The measured concentrations varied between 10 and 100 ng/μl.

## RNA sequencing and analysis

The RNA samples were sequenced by Cambridge Genomic Services (CGS) using their low-input RNA assay. Raw fastq files were submitted to EMBL-EBI ArrayExpress with accession number E-MTAB-13503. Single-end, 75 bp length RNA sequence quality control was performed using fastqc (version 0.11.9)[50], TrimGalore (version 0.6.6)[51], then aligned to the Rattus Norvegicus genome (Rnor_6.0) using STAR (version 2.6.1 d)[52], and gene counts were generated using HTSeq (subread version 2.0.1)[53]. The above analysis was performed by CGS. The summary of the mapping statistics and number of genes identified for each library is given in the respective table (Supplementary Data 1).

Differential expression (DE) analysis was mainly performed using the R software (version 4.2.3) DEseq2 (version 1.38.3)[54] pipeline. All possible pairwise comparisons were performed. Normalised counts for fully differentially expressed genes (DEGs) and the lists of paired DEGs can be found in Supplementary Tables 2–11 under https://github.com/xz289/Kreysing_Franze. DESeq2 generated an individual gene counts plot, which was normalised using the median of ratios method. The counts were divided by sample-specific size factors determined by the median ratio of gene counts relative to the geometric mean per gene. The normalised counts were then log2-transformed. Furthermore, for each comparison, a principal component analysis (PCA) plot and a volcano plot were generated.

 

## Immunocytochemistry of CRISPR KD and TTR KD neurons

To fix samples, half the media was removed and replaced with 4% PFA and 15% sucrose for 30 min. Samples were washed 9 times with PBS. Samples were permeabilised for 5 min in 0.1% Triton X. They were washed 3 times with 0.0001% Triton X in PBS and subsequently blocked with 5% serum, 0.0001% Triton in PBS (blocking solution) for 45 min at room temperature. Afterwards, the samples were incubated with the primary antibody overnight at 4 °C in blocking solution and washed 5 times with 0.0001% Triton in PBS. This was followed by incubation with the secondary antibody for 45 min at room temperature in blocking solution. The samples were washed 5 times in PBS with 0.0001% Triton X. If stained for the nucleus, the samples were then incubated with DAPI (Merck, D5942, working concentration: 0.1 µl/ml) in PBS with 0.0001% Triton X for 20 min. Finally, the samples were washed 5 times in PBS and mounted with 9 µl Fluromount-G Mounting Medium (Thermo Fisher Scientific #00-4958-02) and an 18 mm coverslip. The primary antibodies used were TTR (Thermo Fisher, PA5-20742, chicken, 1:400), Piezo1 (proteintech, 28511-1-AP, rabbit, 1:50), and β-tubulin (synaptic systems, #302304, guinea pig, 1:500). The secondary antibodies were goat anti-chicken Alexa Fluor Plus 405(Thermo Fisher, A48260, 1:500), donkey anti-guinea pig 647 (Jackson Immuno Research, 706-605-148, 1:500), and donkey anti-rabbit Alexa Fluor 488 (Thermo Fisher, A-21206, 1:500) Images were taken using a Leica TCS SP8 confocal microscope with a 63× oil objective (NA = 1.4).

Quantitative analysis of the immunostainings was performed using the total intensity values of each channel. The Piezo1 signal was most abundant in the somata. For the Piezo1 KD cells and the control cells, Piezo1 signals were normalised by dividing by the DAPI signal (Supplementary Fig. 6). For TTR KD cells and the control, TTR and tubulin were quantified (Supplementary Fig. 15). TTR was normalised by dividing by the tubulin signal because TTR was strongly expressed along the neurites.

## TOPO TA cloning and DNA sequencing of CRISPR-edited neurons

Genomic DNA from CRISPR-edited primary hippocampal cultures and CTRL cultures was prepared using the Qiagen DNeasy Blood and Tissue Kit (Cat # 69504). For analysis of CRISPR target sites, PCR primers were designed to amplify the targets of one guide each for Piezo1 KD1 (guide #7, see "Methods" section on Piezo1 guides) and KD2 (guide #8, see "Methods" section on Piezo1 guides) and the targets of TTR KD guides #1 and #2 (see "Methods" section on TTR guides) using the All Taq Master Mix Kit Qiagen (Cat # 203144). PCR reactions were performed on a Biorad T100 thermal cycler as follows 2 min 95 °C denaturation; 40 cycles: 5 s 95 °C, 15 s 55 °C, 10 s 72 °C; and final extension 5 min 72 °C. PCR amplicons were gel-purified using the NEB Monarch DNA Gel Extraction Kit (Cat # T1020S) and cloned into either the PCRII Dual Promoter TOPO vector using the Invitrogen TOPO TA Cloning Kit (Cat # 45-0640) or the PCR 2.1 TOPO vector using the TOPO TA Cloning Kit (Cat # 45-0641). Sanger sequencing was performed using the M13F primer. Sequence analysis was performed in a custom-written Python script using the Bio package.

## Xenopus laevis rearing

Xenopus laevis embryos were obtained through in vitro fertilisation and reared in 0.1 x Marc's Modified Ringer's solution (MMR, see Supplementary Methods) at 14–18 °C until they reached the desired developmental stages for the experiments. The embryos were staged according to ref. 55. All embryos used in this study were below embryonic stage 45.

## Xenopus laevis brain exposure and transglutaminase treatment

Xenopus laevis embryos were prepared for TG treatment and atomic force microscopy (AFM) measurements as described previously[14].

Xenopus laevis embryos were anaesthetised using exposed brain medium (EBM, see Supplementary Methods) at stage 33 for transglutaminase experiments or at stage 37–38 for stiffness maps of healthy brain tissue. The embryos were immobilised in a silicone-filled (SYLGARD™ 184 Silicone Elastomer, cured) Petri dish using bent 0.2 mm minutien pins (Austerlitz). The skin and eye primordium were removed from one side, and one of the brain hemispheres was exposed from the dorsal to ventral midline, and from the hindbrain to the telencephalon using 0.1/0.15 mm minutien pins (Austerlitz) in pinholders. The embryos that were exposed were used for subsequent experiments. To increase the stiffness of brain tissue, 0.5 mg/ml (corresponding to 1 unit per 581 µl) TG (Sigma, T5398) was added to the EBM. Control embryos were kept in EBM. Both conditions were kept at 20–21 °C overnight until they reached stage 37–38.

## Xenopus laevis brain AFM measurements

For AFM stiffness mapping, embryos were prepared as described above. Stiffness maps of healthy brains were obtained directly after embryo mounting. For measuring the effect of Transglutaminase treatment on brain stiffness, immobilised embryos, which had their brains exposed, were then incubated in TG or left in a control solution for at least two hours prior to measurement.

The spring constant (k) of tipless silicon cantilevers (Arrow-TL1, Nanoworld) was determined using the thermal noise method. Cantilevers with k values between 0.02 and 0.05 N/m were selected for the experiments. Polystyrene beads with a radius of 37 µm (microParticles GmbH) were glued to the cantilevers. Subsequently, the cantilevers were mounted on a CellHesion-200AFM or CellHesion-300 head (JPK instruments). Force measurements and data analysis based on the Hertz model were done using custom scripts previously described[14,15].

## Xenopus laevis brain FM1-43 imaging and line profile analysis

First, dishes for mounting the embryos for imaging were prepared. For each sample, two adhesive stationary hole reinforcement rings were glued as a stack in the centre of a 35 mm ibidi dish (ibidi #81218-200). FM1-43 (Thermo Fisher Scientific T3163) stock solution was prepared at 1 mg/ml in distilled water. For live staining of embryonic brains, the solution was dissolved at a ratio of 1:200 in EBM. The embryos were incubated at room temperature for 5 min, washed 3 times, and then placed brain-side down in the centre of the adhesive rings in the pre-prepared dishes. A coverslip was then placed on top of the rings to prevent any movement of the sample.

The embryos were imaged using a spinning disk confocal microscope (Leica SP8) with a DS RED filter set and a 10× objective (NA = 0.32). The images were analysed using FIJI. The confocal stack was limited to the in-focus slides and averaged. In order to quantify these signals, we measured the fluorescent intensity in a 100-pixel-wide stripe across the area corresponding to the supraoptic tract (Fig. 4c). The maximum and mean intensity were determined for each line profile, and the ratio maximum/mean intensity was calculated. This ratio was compared between the different conditions using a Mann–Whitney test.

## Correlation of stiffness maps with synaptic density of Xenopus laevis brain tissue

In vivo AFM stiffness maps were recorded as described above. For synaptic staining, the embryos were prepared as described above and imaged on an upright epifluorescence microscope (Thunder DM6B, Leica), equipped with a 20× water immersion objective (HC APO L 20×/0.50). Brain outlines were manually annotated based on brightfield images and exported as .txt files for further analysis. Subsequent data processing was carried out using the BrainFusion Python package, available on GitHub (https://github.com/nik-liegroup/BrainFusion). Fluorescence image channels were binned by averaging over 15 × 15

pixel blocks, then normalised to the maximum intensity within each channel. The processed images were saved in 8-bit format.

To average brain shapes across both AFM and FM1-43 image datasets, brain outlines were uniformly resampled to 150 equally spaced points. An initial affine alignment was performed by fitting an ellipse to each outline and translating the contours to a common coordinate centre. The ellipses were then rotated and scaled anisotropically, based on their major and minor axes.

The same affine transformations were applied to the corresponding image and AFM grid coordinates to maintain spatial consistency. To compute an average contour, each resampled outline was circularly shifted to a common nearest-neighbour starting point and oriented in a clockwise direction. Mean x- and y-coordinates were then calculated at each resampled position to generate the averaged brain shape.

To align individual contours with the averaged shape, dynamic time warping was used to establish a pointwise correspondence between each original outline and the mean contour. When multiple points on the original contour mapped to the same location on the average, their coordinates were averaged to define a single matched point.

Next, planar transformations were modelled using linear radial basis functions, using the matched contour points as control points. These transformations were then applied to deform either the original pixel coordinates or the AFM measurement coordinates into the common averaged shape space.

Once mapped, the deformed data points were interpolated onto a regular grid matching the resolution of the original pixel or AFM grid. The final averaged map was generated by calculating the median intensity or stiffness value at each grid point.

To compare FM1-43 and AFM data, the averaged brain outlines from each modality were aligned and transformed into a shared reference shape. Due to differences in spatial resolution, FM1-43 intensities were averaged within concentric circles centred on each AFM grid point. The radius of each circle was individually optimised to maximise spatial coverage while preventing overlap.

To assess the spatial correspondence between the two datasets, pointwise symmetric correlation analysis was performed by calculating the Pearson correlation coefficient. For categorical comparison, data points were classified as either soft or stiff (AFM) and as low or high intensity (FM1-43), using the 60th percentile of each distribution as the threshold.

Joint distributions of these classifications were quantified by counting the number of points falling into each category combination. Differences in these counts $n$(AFMFM143) were assessed using Fisher's exact test. The odds ratio, indicating how much more likely stiff regions were to occur in areas of high FM1-43 intensity compared to low-intensity areas, was defined as:

$$O = \frac{\frac{n_{stiff,high}}{n_{soft,high}}}{\frac{n_{stiff,low}}{n_{soft,low}}} \qquad (2)$$

Conditional probabilities were calculated by normalising the counts, providing the relative likelihood of each category within the dataset.

Conditional probabilities (normalised counts) were calculated as:

$$P(AFM|FM143) = \frac{n(AFM \wedge FM143)}{n(FM143)} \qquad (3)$$

## Immunohistochemistry of *Xenopus laevis* brains

For whole-brain immunostaining, embryos were prepared as described above. The exposed embryos were then incubated in TG or left in a control solution overnight. At stage 37–38, embryos were fixed in 4% PFA in PBS for 1 h, followed by two washes in PBS. The brains were then dissected out of the embryos in PBS with 0.1% Triton X, using 0.1/0.15 mm minutien pins (Austerlitz) in pinholders. The brains were washed for 30 min in PBT (PBS with 0.2% BSA and 0.1% Triton X). This was followed by blocking in PBT with 10% donkey serum for 45 min. Brains were incubated with a primary mouse anti-acetylated-tubulin antibody (Sigma, T6793, 1:300 in PBT) at 4 °C overnight. The brains were then washed 3 times in PBT. Brains were then incubated with a secondary donkey anti-mouse antibody (Invitrogen, A10037, 1:300) in PBT with 10% donkey serum for 2 h at room temperature. The brains were then washed three times in PBS and incubated with phalloidin 488 (1:200) and DAPI (1:10,000) in PBS. Finally, they were washed 5 times with PBS and mounted in ibidi dishes as described in the FM1-43 imaging section. The embryos were imaged using a confocal microscope (Leica SP8) and a 40× oil-immersion objective (NA = 1.3). The images were processed using FIJI, creating a maximum intensity projection of all slices containing the axonal tract signals. The acetylated tubulin staining reveals the location of axon tracts.

## Reporting summary

Further information on research design is available in the Nature Portfolio Reporting Summary linked to this article.

## Data availability

The datasets generated and/or analysed for this study are available as follows. The raw RNA data is deposited in the BioStudies database with accession E-MTAB-13503. Processed RNA data is deposited in Zenodo [https://doi.org/10.5281/zenodo.12742961]. The remaining data is deposited in Zenodo [https://doi.org/10.5281/zenodo.16910231]. Source data are provided with this paper.

## Code availability

Code used for data analysis in the study is available as follows: Synapse density analysis[56]: https://github.com/lemur01/synpasestat. https://doi.org/10.5281/zenodo.17152358 Patch-clamp analysis[57]: https://github.com/erkannt/patchclamp-analysis https://doi.org/10.5281/zenodo.17154457 Calcium imaging analysis[58]: https://gitlab.com/rknt/cell-activity-from-calcium-imaging. https://doi.org/10.5281/zenodo.17154501 AFM data analysis[59]: https://github.com/FranzeLab/AFM-data-analysis-and-processing/tree/Batchforce_1.1/Batchforce. https://doi.org/10.5281/zenodo.17154037 Correlating AFM and fluorescent maps[60]: https://github.com/FranzeLab/BrainFusion/tree/v0.1-beta https://doi.org/10.5281/zenodo.17157938 Sholl analysis[61]: https://github.com/erkannt/automated-sholl-analysis https://doi.org/10.5281/zenodo.17154451.

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

## Acknowledgements

We would like to thank Prof. Dr. Christine Holt for discussions and advice, and Liz Williams, Rachel McKeown, Kim Evans, and Julia Becker for preparing hydrogels and mental support. We acknowledge funding by the Wellcome Trust (PhD studentship 222280/Z/20/Z to S.M.), Cambridge Trust (scholarship to S.M.), EPSRC (research grants EP/R025398/1 and EP/Y008715/1 to L.M.), MRC (programme grant MR/Y014537/1 to R.T.K.), BBSRC (research grant BB/N006402/1 to R.T.K. and K.F.), the European Research Council (Consolidator Award 772426 MECHEMGUI and Synergy Grant 101118729 UNFOLD to K.F.), the German Research Foundation (DFG) (projects 460333672 CRC1540 EBM and 270949263 GRK2162 to K.F.), and the Alexander von Humboldt Foundation (Alexander von Humboldt Professorship to K.F.).

## Author contributions

E.K., H.G., R.T.K. and K.F. designed the study. H.G.: performed the immunostaining of synapses and patch-clamp experiments. E.K. developed the CRISPR-Cas9 protocols and performed the calcium imaging experiments. E.K. & K.M. performed the RNA-sequencing experiments and WB experiments. E.K. analysed the WBs. E.K. & S.M. carried out the *Xenopus laevis* TG stiffening and imaging. S.V.-S and A.W. performed the AFM measurements. L.M. wrote the code to analyse the synapse density immunostaining. E.K. & D.H. wrote the calcium imaging analysis pipeline, the immunostaining quantification code for KD cells, the *Xenopus laevis* synapse analysis code, and the WB quantification code. K.M. performed the TOPO cloning. N.G. implemented the analysis pipeline for correlating stiffness maps with synaptic staining. E.K. wrote the code to analyse the DNA sequences. X.Z. performed the RNA-sequencing analysis. E.K. performed the pattern analysis of the RNA-sequencing results. D.H., H.G., R.H. & R.T.K. performed the patch-clamp data analysis. T.B. cultured neurons for revision experiments and contributed to image analysis. A.D. supported neuronal cell culture during lockdown. E.K.P. prepared hydrogels for synapse immunostaining experiments. E.K. and K.F. wrote the manuscript, with contributions from all co-authors. R.T.K. and K.F. supervised the project.

## Competing interests

The authors declare no competing interests.
