## [Transparent Peer Review File · Nature Communications]

Environmental stiffness regulates neuronal maturation via Piezo1-mediated transthyretin activity

Corresponding Author: Professor Kristian Franze

Version 0:

Reviewer comments:

Reviewer #1

(Remarks to the Author)

Kreysing et al report that electrical maturation in primary neurons derived from rat embryos is impacted by substrate stiffness and that this is mediated by Piezo1 through TTR activity. Overall, this is a solid piece of showcasing some very nice imaging and patch clamp work. Below, I outline my central comments:

1. These findings build on many previous papers, from this group and others, that axonal growth and guidance, neuronal activity, etc are all impacted by stiffness. Indeed, many papers have shown that cells of the nervous system adopt more in vivo like phenotypes on soft compared to stiff substrates. It has also been previously reported that Piezo ion channels play important roles in these reported stiffness sensing mechanisms. Here, the authors report a similar finding in neuronal maturation, but with a focus on electrical maturation. They link this to TTR downstream of Piezo1, but the general story is very similar to those previously reported. Indeed, when looking for a mechanism to explain increased neuronal maturation on softer surfaces, they look only at Piezo1, suggesting that there was very little new to discover here in terms of the mechanism. Thus, the novelty here is not that substrate stiffness impacts neurons and that this is mediated through Piezo1. Instead, the main finding is that softness impacts the maturation of electrical activity in neurons and that a downstream effector of Piezo1 in this context is TTR.
2. Previous work on stiffness sensing in the CNS has linked (e.g.) axonal guidance with differences in stiffness in the brain, suggesting that axons follow stiffness gradients. Here, the authors link stiffness to neuronal maturation, but the physiological relevance of the soft and stiff surfaces is less clear. The authors mention that the soft and stiff surfaces match those of in vivo measurements of soft and stiff brain regions, but the relevance of these brain regions to neuronal maturation is unclear. Given what we know about the role of softness in neuronal growth and maturation, it would make sense that softness, particularly if it matches the softness of brain regions where this maturation takes place, would promote maturation. But, what then do the stiff surfaces represent? For example, are there regions of the brain that are stiffer where neurons do not mature? Without some sort of link to the physiological situation, the stiff surfaces feel a bit just like the 'wrong' culture conditions. Then, when the data is considered in this context, it's not that surprising that neurons would not mature when cultured on the 'wrong' surfaces.
3. In figure 1, the authors show data in support of their claims that neurons are more mature on soft compared to stiff surfaces. These cells are maintained on these surfaces for up to two weeks. For many cell types, substrate stiffness impacts cell morphology and proliferation. Cell morphology and growth are strongly associated with many cellular processes in many types of cells and could impact maturation. Thus, to convincingly link cellular responses to substrate stiffness, it would be important to consider these factors, at least to some extent. The authors show 'cell capacitance' in extended figure 2 and claim that as increases in capacitance were similar on soft and stiff surfaces, neuronal growth is independent of substrate stiffness. However, measurements of capacitance show a large amount of variability, particularly at later time points, and do not give an indication of changes in morphology. More precise assays to measure proliferation and images showing cultures more globally (rather than just single neurons) would be helpful here.
4. Along the same lines as my previous comment, one of the authors' findings is that maturation (generation of action potentials, e.g.) is delayed in cells grown on stiff surfaces. However, it's already known that various processes such as growth are also impacted by substrate stiffness. Is it possible to uncouple the effect of stiffness on maturation from the effect

of stiffness on growth? That is, can the authors findings on delayed maturation on stiff substrates simply be explained by delayed growth? Similarly, does the Piezo1 KD have any effect on growth on the stiff substrates?

5. In the experiments in which the authors use transglutaminase to stiffen xenopus brains, the authors claim that delayed maturation could not be attributed to alterations in axon growth. But stiffness is known to impact axon growth. The authors show an image of staining for acetylated tubulin to back up their claim, but there is only 1 image each of the control and treatment conditions, which don't look that similar. Perhaps the timing of the axonal growth in xenopus supports their claim? If not, how do they explain the lack of impact on axonal growth in their experiments? Can they provide more evidence than just this single image that axonal growth is not impaired? The data in Fig. 4l is not all that convincing in support of this either because (particularly in the control) the n numbers are low and the spread of data so wide.

Reviewer #2

(Remarks to the Author)

This interesting study investigates how the stiffness of the environment affects neuronal maturation. The data are interpreted in a way that suggests that hippocampal cultured neurons grown on stiffer substrates mature more slowly, exhibiting delayed synapse formation (measured via IHC) and electrical activity (using electrophysiology). The authors claim, based on Ca^{++} imaging data and CRISPR mediated knockdown, that the mechanosensitive ion channel Piezo1 mediates this effect, suppressing the expression of transthyretin (TTR) on stiff substrates. Thus, TTR's downstream regulation of synaptic receptor expression is altered, hindering electrical maturation. Experiments in *Xenopus laevis* embryos confirm that increased brain tissue stiffness delays synapse formation *in vivo*. These findings reveal that environmental stiffness could be a crucial regulator of neuronal development, with implications for understanding brain circuitry and neurodevelopmental disorders. If there interpretation is correct, this provides a major advance towards understanding how physical properties of the developing brain play a role in the wiring of the nervous system. However, the data could be interpreted in alternate ways.

Overall, the experiments seem carefully done, but may or may not support their main hypothesis and proposed pathway of action. There are several statistical details missing and some questions about how the data is interpreted as outlined below.

Major Critiques

1. The statistical details are lacking, for example in figure 1e, although they describe a 2-way ANOVA, there are no details about the number of cells analyzed or how many culture dishes they came from. Similarly, in figure 1g, although n is stated, it is not clear how these percents were calculated, or how many different culture dishes were tested per group. It should be able to obtain error bars here and do some statistics. Also it is not clear what makes a cell count as active. Is it a single AP observed once? Why not just quantify spike number across cells and generate FI curves, as is the standard in the field?
2. The legend on Fig 2 does not really indicate the finding described in the figure which is that Piezo KD rescues the effects of a stiff substrate on neuronal activity. likewise, the statistical comparisons are not the most relevant ones. I'd also like to see comparisons between soft and hard substrates across conditions.
3. It is unclear why they switched from electrophysiology to imaging between figure 1 and 2. If they want to do this, they first need to show that stiffness has an effect on Ca^{++} signals in the same way it does on APs. There's also a lot of variability, for example they say that at 5DIV there is no activity across any of the conditions, but it is clear that there's a large proportion of KD1 cells in the soft substrate that are very active. Finally, why is there a difference between KD1 and KD2 at DIV7?
4. I would really like to see some ephys done with the Piezo CDs, or at least more convincing evidence that the Ca signal is indicating the same thing. The TTX experiment is not convincing. Also, Piezo might be having an effect on AP generation that is different from synaptic inputs, and it is difficult to tell from the Ca data alone.
5. Figure 3. While there seems to be an effect of substrate on TTR, AMPA and GABA receptor levels, I don't see evidence that Piezo1 is responsible for differences in the glutamate and GABA receptors, not that TTR is responsible for regulating these levels. The relationship could be purely correlative.
6. Figure 4, the observation is interesting, but again could be correlative. It is known that the ECM limits the amount of plasticity and synaptogenesis in the developing CNS, for example via formation of PNNs. The TG treatment may be limiting synaptogenesis via effects on PNNs (for example by limiting effect of MMPs), and not necessarily via the stiff-Piezo pathway.

Conclusions

In sum, it is hard to tell whether the authors have found a series of correlated, but not necessarily linked observations, or whether they have tapped into an interesting pathway. At minimum the authors should acknowledge the limitations of their mixed approach, better justify their experimental decisions, provide some ground-truth measurements for their Ca^{++} imaging data, and rethink their statistical comparisons as well as provide more transparency for their statistical comparisons.

Reviewer #3

(Remarks to the Author)

Environmental stiffness regulates neuronal maturation via Piezo1-mediated TTR activity

Here, Kreysing and colleagues investigate the effect of substrate stiffness on the formation and maturation of synapses and intrinsic excitability. First, they show that compared to hippocampal neurons grown on the soft substrate, cultures grown on the stiffer substrate have a lower density of synapses (glutamate and GABA), decreased Na⁺ current densities, and a lower fraction of neurons that can fire action potentials. Through CRISPR/CAS9 editing it is established that the stiff substrate elicits these effects through activating Piezo1 channels. Next, they carry out RNA sequencing of Control and Piezo1 KD cultures grown on both soft and stiff substrates and find that Piezo1 inhibits synapse maturation and overall neural activity by inhibiting the expression of transthyretin. Transthyretin is known to be involved in upregulation of GABA and NMDA receptors. Altogether, these data reveal a pathway in which the stiff substrate activates Piezo1, which inhibits transthyretin, thereby downregulating (synaptic) GABA and glutamate receptors. Because the soft substrate does not activate Piezo1, transthyretin is not inhibited, and so can carry out its role in the normal upregulation of these synaptic receptors during development. Finally, this effect of stiff substrate is studied and confirmed *in vivo*, in the developing *Xenopus* tadpole retinotectal projection. Overall, this study is very clearly written and logical. I enjoyed reading it. The study makes excellent use of the hippocampal culture system which allows for precise manipulation of the substrate, and for carrying out whole cell electrophysiological recordings and calcium imaging. The study also includes a sophisticated approach for knocking down Piezo using CRISPR/Cas9 system, and RNAseq to identify TTR. These findings and the overall pathway identified should be of great interest to the overall developmental neuroscience community and should make an important and interesting addition to developmental textbooks. This reviewer's main question is whether there is sufficient evidence to show that the decrease in synapse density of neurons cultured on stiff substrate is not due to impaired dendritic outgrowth and so less chance for axon/dendrite interaction (See first point below). This is important since it is well known that substrates in culture affect the rate of axon and dendrite outgrowth. Also, the final model proposed by the authors does not account for how piezo1 activation is inhibiting intrinsic excitability, only synaptic receptors. The following are suggestions on how the manuscript could be further improved.

Results

In this study, it is determined that neuronal cultures grown on a stiff substrate have a lower density of synapses compared to those grown on a soft substrate. The interpretation is that that stiff substrate is inhibiting synapse formation/maturation. However, given that rates of axon and dendrite outgrowth in primary neuronal cultures are known to be highly dependent on substrate, it is possible that the decrease in synapse density observed on stiff substrate could be due to overall slower growth and therefore fewer axon/dendrite interactions. The authors do report that axon dynamics look normal on experimentally stiffened substrate *in vivo* (Extended Data Figure 12), so probably this holds true in culture system. However, there seems to be no information on dendrite formation/pattern of arborization on the 2 different substrates. The authors do note in the electrophysiology section of the Results that the capacitance of neurons on the stiff and soft substrates are not different. It seems like this is interpreted to mean that the stiff substrate is not eliciting an effect on overall morphology (total membrane area). But voltage clamp is limited to the soma and very proximal dendrite (i.e. the amplifier does not have the ability to clamp out beyond the very proximal portion of the dendrite). Thus, capacitance measurements from the soma cannot reflect dendritic area/complexity. Therefore, dendritic arbor complexity on soft versus stiff substrate should be measured, or at very least it should be stated in Results that impaired/compromised dendrite arbor formation on stiff substrate cannot be ruled out as contributing to the decrease in synapse density observed for neurons cultured on stiff substrate.

Results and Figure 1e (I-V plots): How was the current density measured? In other words, what values were used to determine statistical significance (i.e. peak Na⁺ current density)? It is stated that a 2-way ANOVA test was carried out, but on exactly what data/values? It is a tad surprising that there is a statistically significant difference in Na⁺ current densities between stiff and soft substrates at DIV10 and DIV 14.

Results/ Figure 1f:

Regarding the % active cells versus DIV: How many action potentials would a neuron need to fire during the recording to be considered "active"? The Methods states that neurons were recorded up to 10 minutes. Does this mean that a given neuron was recorded up until an action potential was observed? The criteria used to determine whether a neuron was "active" or not should be more clearly explained at least in the Methods section. Also, it would be informative and interesting to see plots of the frequency of firing, although this may not be possible if the neurons were recorded for varying lengths of time ("recorded up to 10 minutes").

Results/ Figure 1g

How many action potentials must a given neuron need to fire to be considered active? What was the criteria for an event to be counted as an action potential?

Also, it would be great to display these data as F-I curves (frequency or number of action potentials fired during the 200ms step as a function of the amount of current injected into soma). This way, we can see how spiky the neurons are – a measure of intrinsic excitability – not just whether they are able to spike in response to somatic current injection.

Results/ Figure 2

Next, the authors ask whether piezo1 underlies the effect of stiff substrate on neuronal activity. For this, the activity of piezo1 kd neurons on stiff and soft substrate is measured via somatic calcium imaging instead of measuring IE via whole cell electrophysiology. What is the rationale for switching from the electrophysiological approaches for measuring neural activity to calcium imaging? To improve logical flow of the experiments, perhaps the rationale for the different approaches for measuring neuronal activity (electrophys Fig. 1 versus calcium imaging Fig. 2) could be explained in the manuscript.

Figure 2: The heading of Figure 2: Shouldn't this be "Piezo1 delays electrical maturation of neurons on stiff substrates"?

Results/ Figure 3

In the section "Piezo1 regulates substrate stiffness-dependent GABAR and GLUR1 expression via Transthyretin":

Second paragraph, last sentence of this section states that Piezo1 KD led to similarly high TTR expression and electrical activity on soft and stiff substrates and references Fig. 3b.

However, Figure 3B only shows TTR expression levels. It does not show electrical activity.

Figure 3f does show electrical activity of TTR KD on soft tissue versus Control neurons on stiff substrate, but it seems this panel is not referenced in the Results section. This needs to be corrected.

Figure 3c: (western blots of TTR, GluR1, GABAR, and NMDAR)

Fig. 3c (left): What is "TPS"? It is not mentioned in the legend nor in the Results section. Please define this at least in the legend and explain why it is included in this blot.

Fig3c (middle and left): Please define tubulin in the legend and why it is included in the western blot.

Fig. 3d: it would be comforting to see the signal ratio for tubulin included here, or at least mention its ratio in the legend (since it is probably a control).

Discussion

The model (Figure 3e) does not include a proposed mechanism on how the stiff substrate ultimately impairs maturation of intrinsic excitability. Perhaps this could be acknowledged and/or a theory could be proposed. For instance, could the decrease in synaptic strength somehow hinder that maturation of IE? Alternatively, is it known whether TTR has a direct effect on Na⁺ channels?

Methods:

Electrophysiology

For the voltage-step protocol, the baseline voltage at which neurons were held at and brought back to after each voltage step needs to be indicated.

Why was lucifer yellow included in the internal recording solution?

The Methods states that the neuron's membrane resistance was measured. However, membrane resistances do not appear to be reported. Given that the ability of a neuron to fire action potentials is shaped by its membrane resistance, the average membrane resistances for neurons on soft and stiff substrates is important to know and should be reported. The resting membrane potentials – an indicator of overall neuron health – should also be reported if possible.

Version 1:

Reviewer comments:

Reviewer #1

(Remarks to the Author)

The authors have done a good job of addressing my previous comments. In particular, the new data in Figure 5 very nicely addressed one of my central concerns. I have no further comments.

Reviewer #2

(Remarks to the Author)

1. While the new manuscript has been improved with new experiments and clarification of statistical details, there is still some opaqueness that remains. For example, comparing the %active cells in figure 2 (electrophysiology) vs figure 3 (Ca imaging), there are differences in equivalent groups. In fig 2, in soft substrate at DIV7, the percent active is around 50%, while in figure 3, under the same conditions, it is much more lower (~20%). Likewise in the Ca⁺⁺ imaging figure, at 5DIV the soft substrate cells show a substantial amount of activity. The authors say the median value in this condition is 0, but this is deceptive. From what I can see, about 8/21 fields show significant activity, the median is zero because more fields showed no activity, but that's ignoring 1/3 of the data! Also how was cell health assessed in fields with zero activity? Without ground truth experiments, I am unconvinced the Ca⁺⁺ transients necessarily represent the same measure as the electrophysiology, and it concerns me that while some trends are the same, some are different.

2. The added patch clamp studies in the Piezo mutants was a good addition, I would report the values of the ANOVA describes in the text, the area under the curve measurements for the FI curves are not standard in the field.

3. I am also confused by this comment in the rebuttal "We respectfully disagree with the reviewer. The role of TTR in regulating glutamate and GABA receptor expression has been established in multiple prior studies, which we now reference more explicitly in the revised manuscript. And, as acknowledged by the reviewer, we unambiguously show that substrate stiffness regulates TTR via Piezo1. Hence, our data strongly support a mechanistic, not purely correlative, relationship."

There used to be data on GABA and Glu receptors, but that has been removed, so unclear how the authors continue so support that claim, or otherwise I must have missed their point.

4. The authors claim to show in Sup Fig15 to show data where they "knocked down TTR expression in neurons and used calcium imaging to determine the fraction of active cells on soft and stiff substrates at DIV 7", but these data are missing, the referenced figure shows tubular staining, which is also not clear what that was for.

In sum, there are improvements, but some statistical decisions remain opaque, the data is disorganized (eg. why is the Piezo KD data in figure 2 and not in figure 3 where it better fits)? The figures are hard to read with tiny labels (eg. Figure 2). Thus the paper is extremely hard to follow. I suggest an overhaul of figure design, with clearer stats and selection of which data are relevant and which aren't.

Reviewer #3

(Remarks to the Author)

All concerns have been well-addressed. Love the new F-I curves! Great work!

Version 2:

Reviewer comments:

Reviewer #2

(Remarks to the Author)

The authors have addressed most of my concerns, the adjusted statistics and figure redesign have helped with clarity. I still think there's more going on with the Ca⁺⁺ imaging that the authors are overlooking, but I agree that at least the trends are all in the same direction, so we'll go with that.

I also noticed in Extended Figure 2 (sorry that I missed this earlier), with the Sholl analysis, that the authors perform some sort of "area under the curve" analysis again. This is not standard in the field (at least not as far as I know), unclear what the units (um) even tell you since it is measuring intersections, and they should change this to an ANOVA or some other test to compare differences in branch distribution. If they want other quantitative comparisons they could for example compare total dendritic branch length, or total number of branch tips.

Other than that, I think the general conclusions of the paper stand.

Response to Reviewers

We would like to thank the reviewers for their excellent feedback and encouraging comments. Their questions have greatly helped us to improve the manuscript. We were able fully to address all the reviewers' points as detailed below. Changes in the manuscript are **highlighted in yellow**.

Reviewer Comments and Answers to the Comments:

Reviewer #1 (Remarks to the Author):

Kreysing et al report that electrical maturation in primary neurons derived from rat embryos is impacted by substrate stiffness and that this is mediated by Piezo1 through TTR activity. Overall, this is a solid piece of showcasing some very nice imaging and patch clamp work.

We would like to thank the reviewer for this positive assessment of our work.

Below, I outline my central comments:

1. These findings build on many previous papers, from this group and others, that axonal growth and guidance, neuronal activity, etc are all impacted by stiffness. Indeed, many papers have shown that cells of the nervous system adopt more in vivo like phenotypes on soft compared to stiff substrates. It has also been previously reported that Piezo ion channels play important roles in these reported stiffness sensing mechanisms. Here, the authors report a similar finding in neuronal maturation, but with a focus on electrical maturation. They link this to TTR downstream of Piezo1, but the general story is very similar to those previously reported. Indeed, when looking for a mechanism to explain increased neuronal maturation on softer surfaces, they look only at Piezo1, suggesting that there was very little new to discover here in terms of the mechanism. Thus, the novelty here is not that substrate stiffness impacts neurons and that this is mediated through Piezo1. Instead, the main finding is that softness impacts the maturation of electrical activity in neurons and that a downstream effector of Piezo1 in this context is TTR.

We fully agree with the reviewer. Previous studies have established a role for Piezo1 in neuronal development and mechanosensation. Our current work links environmental stiffness and Piezo1 activity specifically to the timing of the electrical maturation in neurons, which is still poorly understood. In addition, we identify TTR as a downstream effector in this process—an interaction that, to our knowledge, has never been previously described and adds further insight into how mechanical signals can influence neuronal excitability during development. This newly identified downstream target of Piezo1 might also be critically involved in many other biological processes in different systems. Furthermore, our *in vivo* experiments add a new facet to the literature on neuronal mechanosensing and on neuronal maturation. In order to better differentiate the presented work from previous studies and emphasize the novelty of the current study, we have modified the abstract, introduction, and the discussion (pages 3, 5, 14).

2. Previous work on stiffness sensing in the CNS has linked (e.g.) axonal guidance with differences in stiffness in the brain, suggesting that axons follow stiffness gradients. Here, the authors link stiffness to neuronal maturation, but the physiological relevance of the soft and stiff surfaces is less clear. The authors mention that the soft and stiff surfaces match those of in vivo measurements of soft and stiff brain regions, but the relevance of these brain regions to neuronal maturation is unclear. Given what we know about the role of

softness in neuronal growth and maturation, it would make sense that softness, particularly if it matches the softness of brain regions where this maturation takes place, would promote maturation. But, what then do the stiff surfaces represent? For example, are there regions of the brain that are stiffer where neurons do not mature? Without some sort of link to the physiological situation, the stiff surfaces feel a bit just like the ‘wrong’ culture conditions. Then, when the data is considered in this context, it’s not that surprising that neurons would not mature when cultured on the ‘wrong’ surfaces.

We would like to thank the reviewer for raising this excellent point, which triggered new experiments to investigate the relationship between tissue stiffness and synapse formation *in vivo*. The developing brain is indeed mechanically heterogeneous. We found a significant negative correlation between brain tissue stiffness and synapse formation, with a four times higher probability for synapses to form in soft areas than in stiff areas. These new data support the idea that stiffer environments delay neuronal maturation - both *in vitro* and *in vivo* - and they show that our stiff substrates *in vitro* are not the ‘wrong culture conditions’ but rather conditions that are also found *in vivo* – with a comparable phenotype. We have added these new experiments to the revised manuscript (new Figure 5b-d), and added an according interpretation to the discussion.

3. In figure 1, the authors show data in support of their claims that neurons are more mature on soft compared to stiff surfaces. These cells are maintained on these surfaces for up to two weeks. For many cell types, substrate stiffness impacts cell morphology and proliferation. Cell morphology and growth are strongly associated with many cellular processes in many types of cells and could impact maturation. Thus, to convincingly link cellular responses to substrate stiffness, it would be important to consider these factors, at least to some extent. The authors show ‘cell capacitance’ in extended figure 2 and claim that as increases in capacitance were similar on soft and stiff surfaces, neuronal growth is independent of substrate stiffness. However, measurements of capacitance show a large amount of variability, particularly at later time points, and do not give an indication of changes in morphology. More precise assays to measure proliferation and images showing cultures more globally (rather than just single neurons) would be helpful here.

This point is addressed together with the next point (see below).

4. Along the same lines as my previous comment, one the of the authors’ findings is that maturation (generation of action potentials, e.g.) is delayed in cells grown on stiff surfaces. However, it’s already known that various processes such as growth are also impacted by substrate stiffness. Is it possible to uncouple the effect of stiffness on maturation from the effect of stiffness on growth? That is, can the authors findings on delayed maturation on stiff substrates simply be explained by delayed growth? Similarly, does the Piezo1 KD have any effect on growth on the stiff substrates?

To address these two related points, we conducted a quantitative assessment of dendritic arbor complexity using Sholl analysis, the standard method in the field. Our results revealed no significant differences between neurons cultured on soft vs. stiff substrates, or between control neurons and cells with depleted Piezo1. These findings have been incorporated into the manuscript as new Extended Data Figure 2. We cultured E18 hippocampal neurons, which are post-mitotic and non-proliferative *in vitro*, thereby ruling out cell division as a source of morphological variation. In addition, to provide a more global view of the cultures beyond individual neurons, we have included lower-magnification brightfield images in Extended Data Figure 9 as suggested by the reviewer. In summary, we did not observe any effects of substrate stiffness or Piezo1 on the

morphology of hippocampal neurons, in agreement with previous studies (e.g., Koch et al, BiophysJ 2012). Hence, we are confident that the delay in neuronal maturation on stiff substrates is not a consequence of differential neurite growth.

5. In the experiments in which the authors use transglutaminase to stiffen xenopus brains, the authors claim that delayed maturation could not be attributed to alterations in axon growth. But stiffness is known to impact axon growth. The authors show an image of staining for acetylated tubulin to back up their claim, but there is only 1 image each of the control and treatment conditions, which don't look that similar. Perhaps the timing of the axonal growth in xenopus supports their claim? If not, how do they explain the lack of impact on axonal growth in their experiments? Can they provide more evidence than just this single image that axonal growth is not impaired? The data in Fig. 4I is not all that convincing in support of this either because (particularly in the control) the n numbers are low and the spread of data so wide.

We agree that tissue stiffness can influence axonal growth and pathfinding. However, the formation of the supraoptic tract, which we investigated here, occurs already prior to the developmental time at which we started the transglutaminase treatment (stage 33–34), as the reviewer correctly speculated. We apologise for not having provided this important piece of information in the initial manuscript. As the timing of the appearance of the supraoptic tract is well-established, we would rather not want to sacrifice extra animals to increase numbers. We have now added this clarification to the main text, along with a reference supporting the timing of supraoptic tract development (page 13).

Reviewer #2 (Remarks to the Author):

*This interesting study investigates how the stiffness of the environment affects neuronal maturation. The data are interpreted in a way that suggests that hippocampal cultured neurons grown on stiffer substrates mature more slowly, exhibiting delayed synapse formation (measured via IHC) and electrical activity (using electrophysiology). The authors claim, based on Ca⁺⁺ imaging data and CRISPR mediated knockdown, that the mechanosensitive ion channel Piezo1 mediates this effect, suppressing the expression of transthyretin (TTR) on stiff substrates. Thus, TTR's downstream regulation of synaptic receptor expression is altered, hindering electrical maturation. Experiments in *Xenopus laevis* embryos confirm that increased brain tissue stiffness delays synapse formation in vivo. These findings reveal that environmental stiffness could be a crucial regulator of neuronal development, with implications for understanding brain circuitry and neurodevelopmental disorders. If there interpretation is correct, this provides a major advance towards understanding how physical properties of the developing brain play a role in the wiring of the nervous system.*

We would like to thank the reviewer for acknowledging the importance of our work.

However, the data could be interpreted in alternate ways. Overall, the experiments seem carefully done, but may or may not support their main hypothesis and proposed pathway of action. There are several statistical details missing and some questions about how the data is interpreted as outlined below.

Major Critiques

1. The statistical details are lacking, for example in figure 1e, although they describe a 2-

way ANOVA, there are no details about the number of cells analyzed or how many culture dishes they came from. Similarly, in figure 1g, although n is stated, it is not clear how these percents were calculated, or how many different future dishes were tested per group. It should be able to obtain error bars here and do some statistics.

We thank the reviewer for pointing out the missing details and apologise for their omission in the original manuscript. We added numbers of cells to each figure panel, replotted figure 1g (now 2j, k) and added a statistical analysis, and provide details on how percentages were calculated in the figure caption now.

Also it is not clear what makes a cell count as active. Is it a single AP observed once? Why not just quantify spike number across cells and generate FI curves, as is the standard in the field?

A cell was considered active if it spontaneously produced at least one action potential during the recording. In the original experiments, cells were recorded for up to 10 minutes to assess spontaneous activity. Recordings were stopped as soon as an action potential was detected.

We thank the reviewer for suggesting to present the evoked AP data as frequency–current (F–I) curves. The figure has been updated accordingly, and we added statistical comparisons of the curves across the different substrates.

2. The legend on Fig 2 does not really indicate the finding described in the figure which is that Piezo KD rescues the effects of a stiff substrate on neuronal activity. likewise, the statistical comparisons are not the most relevant ones. I'd also like to see comparisons between soft and hard substrates across conditions.

We have revised the legend of Figure 2g (now Figure 3g) to more clearly state that Piezo1 knockdown rescues the effect of the stiff substrate on neuronal activity as suggested by the reviewer. The requested comparisons between conditions can be found in Extended Data Figure 14 (they were already included in the first submission, Extended Data Figure 9). The plot strongly supports our finding that Piezo1 activity on stiff substrates slows down the electrical maturation of neurons. We now refer to this alternative data representation also in the caption of the main figure to make it easier to find.

3. It is unclear why they switched from electrophysiology to imaging between figure 1 and 2. If they want to do this, they first need to show that stiffness has an effect on Ca⁺⁺ signals in the same way it does on APs. There's also a lot of variability, for example they say that at 5DIV there is no activity across any of the conditions, but it is clear that there's a large proportion of KD1 cells in the soft substrate that are very active. Finally, why is there a difference between KD1 and KD2 at DIV7?

While patch-clamp recordings provide much more detailed electrophysiological data, the low throughput would not have allowed to collect sufficient data across all six conditions at three different time points. Sticking with patch clamp electrophysiology would simply have rendered the project, which also required skills in engineering approaches, programming, and molecular biology, unfeasible within the duration of a postdoc position. Therefore, we switched to calcium imaging, as the relationship between calcium transients and action potentials is well-established and it enables the analysis of much larger data sets within a reasonable timeframe.

For example, parallel recordings using patch-clamp electrophysiology and calcium imaging have demonstrated a clear correlation between calcium transients and electrophysiological measurements of action potentials. Only at very high firing frequencies, which we did not observe in our patch clamp experiments, calcium peaks begin to merge (e.g., Smetters et al., *Methods* 1999; Lock et al., *Cell Calcium* 2015). Furthermore, our analysis categorizes cells as active or non-active, based on the presence of at least one detectable fluorescence peak. Thus, any potential merging of calcium peaks at high firing rates would not impact our main findings.

We have now added the explanation for the change in methods to the manuscript and incorporated a reference on parallel calcium imaging and patch-clamp recordings (page 10).

Regarding the activity of KD1 cells at DIV5 on soft substrates, the median number of fields of view (FOV) containing active cells is 0. This was initially indicated by a narrow black line, which we appreciate was difficult to distinguish from the graph axis. To make this clearer, we now include the total number of FOVs in the figure, and we have changed width of the median bar to make it more visible. Statistically, this condition is not significantly different from any of the others at DIV5, as indicated in the graph and also in the Extended Data Figure 14.

The reviewer is correct in noting some variability in activity across the different conditions. This variability is expected because we are working with primary neurons in which we induced a knockdown of Piezo1 using CRISPR-Cas9 rather than with neurons exhibiting a stable knockout of the gene. Each knockdown condition was generated using four distinct CRISPR-Cas9 guides, which introduce random cuts around the target sites. The subsequent repair of double-strand breaks is also random, and whether this repair leads to frameshift mutations—and consequently nonsensical amino acid sequences—depends on the error-prone repair process. Additionally, the efficiency of each guide and the probability of frameshift mutations might vary. As a result, each group contains a heterogeneous population of cells and the overall effect on the behaviour of a dish can vary in strength between the different KD conditions. We now explain this in the Methods section (page 36).

4. I would really like to see some ephys done with the Piezo CDs, or at least more convincing evidence that the Ca signal is indicating the same thing. The TTX experiment is not convincing. Also, Piezo might be having an effect on AP generation that is different from synaptic inputs, and it is difficult to tell from the Ca data alone.

To address this point, we conducted new patch-clamp recordings of Piezo1 knockdown (KD) and control neurons cultured on both soft and stiff substrates. To assess neuronal excitability, we measured current densities and recorded evoked action potentials. Consistent with our calcium imaging data, Piezo1 KD neurons displayed high firing rates across both substrate conditions, while control neurons exhibited significantly reduced firing rates on stiff substrates (see new Figure 2I-o). These electrophysiological results support the interpretation that Piezo1 modulates neuronal excitability in a substrate stiffness-dependent manner. While we initially used TTX to confirm that calcium transients are action potential-dependent, the new patch clamp recordings directly show that Piezo1 knockdown neurons generate significantly more action potentials on stiff substrates than control neurons, in strong support of our initial interpretation of the data.

5. Figure 3. While there seems to be an effect of substrate on TTR, AMPA and GABA

receptor lives, I don't see evidence that Piezo1 is responsible for differences in the glutamate and GABA receptors, not that TTR is responsible for regulating these levels. The relationship could be purely correlative.

We respectfully disagree with the reviewer. The role of TTR in regulating glutamate and GABA receptor expression has been established in multiple prior studies, which we now reference more explicitly in the revised manuscript. And, as acknowledged by the reviewer, we unambiguously show that substrate stiffness regulates TTR via Piezo1. Hence, our data strongly support a mechanistic, not purely correlative, relationship. We now emphasise these thoughts and clarify the interpretation of the RNA sequencing results in more detail in Figure 3b (now Fig. 4b). Please also see our reply to Reviewer 3's question about Fig. 3d on pages 11 and 12 of this rebuttal letter.

6. Figure 4, the observation is interesting, but again could be correlative. It is known that the ECM limits the amount of plasticity and synaptogenesis in the developing CNS, for example via formation of PNNs. The TG treatment may be limiting synaptogenesis via effects on PNNs (for example by limiting effect of MMPs), and not necessarily via the stiff-Piezo pathway.

We appreciate the reviewer's comment regarding a potential involvement of perineuronal nets (PNNs) in the regulation of neuronal plasticity and synaptogenesis. While PNNs might principally also regulate neuronal function via changes in the stiffness of the neuronal environment, to the best of our knowledge there are no reports of PNNs in the brains of *Xenopus* embryos at the investigated stages. The only available paper looking into PNNs in early stage anurans identified lectins in brains of *Rhinella yunga* (beaked toad) embryos (Edwards et al., J Comp Neurol, 2021). However, lectins are carbohydrate-binding proteins which should not bind to transglutaminase, as it is not glycosylated. In other species, PNNs form relatively late in development—for example, more than two weeks after birth in mice (Mirzadeh et al., Nat. Metab., 2019). Thus, the cross-linking of mature PNNs by transglutaminase is not very likely a relevant factor at these early embryonic stages in *Xenopus*. However, we cannot fully exclude it and now explicitly discuss PNNs in the revised version of our manuscript (page 14).

We also explored an AFM-based compression-stiffening approach (Koser et al., Nat Neurosci 2016; Barriga et al., Nature 2018) to locally increase the stiffness in the region of the supraoptic tract without directly affecting the chemistry of the tissue (brain tissue stiffens under compression: Pogoda et al., New J Phys 2014). This experiment involved the application of mechanical stress on the supraoptic tract with an AFM cantilever. The maximum effective contact area we could reliably achieve was approximately 50 μm , i.e., on the order of the width of the tract. Hence, the cantilever had to be positioned right above the supraoptic tract with high precision, otherwise the environment of the tract would not get stiffer. Unfortunately, we did not find a way to label the supraoptic tract in the living embryos and visualize it at the relevant stages during the AFM experiments (due to significant imaging limitations at low fluorescence intensities in the setup). After many unsuccessful attempts, we had to give up.

To provide further evidence of an important role of brain tissue stiffness in regulating synapse formation *in vivo*, we now conducted a new set of experiments (new Figure 5b-d). These experiments revealed a very significant negative correlation between local brain stiffness and synapse formation, with soft areas being four times more likely to form synapses than stiff areas. Together with our previous results, these new findings provide

strong support for our hypothesis that increased environmental stiffness delays synaptic maturation, both *in vitro* and *in vivo*.

Conclusions

In sum, it is hard to tell whether the authors have found a series of correlated, but not necessarily linked observations, or whether they have tapped into an interesting pathway. At minimum the authors should acknowledge the limitations of their mixed approach, better justify their experimental decisions, provide some grid-truth measurements for their Ca⁺⁺ imaging data, and rethink their statistical comparisons as well as provide more transparency for their statistical comparisons.

We hope that our new experiments, plots and details on experimental decisions and statistical analyses convincingly show that we have indeed discovered a novel molecular pathway linking tissue stiffness to neuronal maturation – a discovery that likely is highly relevant for many other biological systems.

Reviewer #3 (Remarks to the Author):

Environmental stiffness regulates neuronal maturation via Piezo1-mediated TTR activity

*Here, Kreysing and colleagues investigate the effect of substrate stiffness on the formation and maturation of synapses and intrinsic excitability. First, they show that compared to hippocampal neurons grown on the soft substrate, cultures grown on the stiffer substrate have a lower density of synapses (glutamate and GABA), decreased Na⁺ current densities, and a lower fraction of neurons that can fire action potentials. Through CRISPR/CAS9 editing it is established that the stiff substrate elicits these effects through activating Piezo1 channels. Next, they carry out RNA sequencing of Control and Piezo1 KD cultures grown on both soft and stiff substrates and find that Piezo1 inhibits synapse maturation and overall neural activity by inhibiting the expression of transthyretin. Transthyretin is known to be involved in upregulation of GABA and NMDA receptors. Altogether, these data reveal a pathway in which the stiff substrate activates Piezo1, which inhibits transthyretin, thereby downregulating (synaptic) GABA and glutamate receptors. Because the soft substrate does not activate Piezo1, transthyretin is not inhibited, and so can carry out its role in the normal upregulation of these synaptic receptors during development. Finally, this effect of stiff substrate is studied and confirmed *in vivo*, in the developing *Xenopus* tadpole retinotectal projection.*

Overall, this study is very clearly written and logical. I enjoyed reading it. The study makes excellent use of the hippocampal culture system which allows for precise manipulation of the substrate, and for carrying out whole cell electrophysiological recordings and calcium imaging. The study also includes a sophisticated approach for knocking down Piezo using CRISPR/Cas9 system, and RNAseq to identify TTR. These findings and the overall pathway identified should be of great interest to the overall developmental neuroscience community and should make an important and interesting addition to developmental textbooks.

We would like to thank the reviewer for this very positive assessment of our study.

This reviewer's main question is whether there is sufficient evidence to show that the decrease in synapse density of neurons cultured on stiff substrate is not due to impaired dendritic outgrowth and so less chance for axon/dendrite interaction (See first point

below). This is important since it is well known that substrates in culture affect the rate of axon and dendrite outgrowth.

As we explain in more detail below, the sensitivity of neurite growth to substrate stiffness is neuronal cell type-dependent. Previous work showed that neurite growth in hippocampal neurons, which were used in the current study, actually does not depend on substrate stiffness (e.g., Koch et al., Biophys J 2012). Our new experiments assessing axonal and dendritic morphologies confirm these findings (new Extended Data Figure 2), excluding an effect of differential dendritic growth on synapse densities (see below).

Also, the final model proposed by the authors does not account for how piezo1 activation is inhibiting intrinsic excitability, only synaptic receptors.

We thank the reviewer for this comment. We now speculate on a potential mechanism linking Piezo1 activity to intrinsic excitability in the discussion (page 13-14, please also see our response to a later comment) and added this hypothesis to Fig. 4e.

The following are suggestions on how the manuscript could be further improved.

Results

In this study, it is determined that neuronal cultures grown on a stiff substrate have a lower density of synapses compared to those grown on a soft substrate. The interpretation is that that stiff substrate is inhibiting synapse formation/maturation. However, given that rates of axon and dendrite outgrowth in primary neuronal cultures are known to be highly dependent on substrate, it is possible that that the decrease in synapse density observed on stiff substrate could be due to overall slower growth and therefore fewer axon/dendrite interactions. The authors do report that axon dynamics look normal on experimentally stiffened substrate in vivo (Extended Data Figure 12), so probably this holds true in culture system. However, there seems to be no information on dendrite formation/pattern of arborization on the 2 different substrates. The authors do note in the electrophysiology section of the Results that the capacitance of neurons on the stiff and soft substrates are not different. It seems like this is interpreted to mean that the stiff substrate is not eliciting an effect on overall morphology (total membrane area). But voltage clamp is limited to the soma and very proximal dendrite (i.e. the amplifier does not have the ability to clamp out beyond the very proximal portion of the dendrite). Thus, capacitance measurements from the soma cannot reflect dendritic area/complexity. Therefore, dendritic arbor complexity on soft versus stiff substrate should be measured, or at very least it should be stated in Results that impaired/compromised dendrite arbor formation on stiff substrate cannot be ruled out as contributing to the decrease in synapse density observed for neurons cultured on stiff substrate.

We would like to thank the reviewer for raising this important point. To address the possibility of differences in dendritic arbor formation, we performed Sholl analysis on neurons cultured on soft and stiff substrates for control as well as Piezo1 knockdown neurons. The analysis revealed no significant differences in dendritic complexity across conditions. These results are now included in the manuscript as new Extended Data Figure 2.

Results and Figure 1e (I-V plots): How was the current density measured? In other words, what values were used to determine statistical significance (i.e. peak Na⁺ current density)? It is stated that a 2-way ANOVA test was carried out, but on exactly what

data/values? It is a tad surprising that there is a statistically significant difference in Na⁺ current densities between stiff and soft substrates at DIV10 and DIV 14.

We apologize for the lack of clarity in our analysis of the ion current density data.

Following the reviewer's comment and internal discussions, we recognized that our initial statistical tests were not the most appropriate for addressing substrate-specific differences in maturation. Since our goal was to compare maturation over time between stiffnesses, we reanalysed the data by plotting and comparing current densities as a function of cell age (DIV) (new Fig. 2). Two-way ANOVAs were then performed separately for sodium and potassium currents, with cell age and substrate stiffness as factors.

We quantified peak sodium current densities at each time point, and potassium current densities were measured at +40 mV. Cell age had a significant effect on both currents, consistent with developmental changes (not shown in the manuscript). A significant effect of substrate stiffness was observed for sodium current densities. Using posthoc analysis, we indeed don't find any significant differences for specific days any longer. We thank the reviewer for raising this concern and updated the figure (now Fig. 2 a-g) and the corresponding text in the Results section accordingly.

Results/ Figure 1f:

Regarding the % active cells versus DIV: How many action potentials would a neuron need to fire during the recording to be considered "active"? The Methods states that neurons were recorded up to 10 minutes. Does this mean that a given neuron was recorded up until an action potential was observed? The criteria used to determine whether a neuron was "active" or not should be more clearly explained at least in the Methods section. Also, it would be informative and interesting to see plots of the frequency of firing, although this may not be possible if the neurons were recorded for varying lengths of time ("recorded up to 10 minutes").

Cells were indeed recorded for up to 10 minutes, and the recording was ended earlier if an action potential was observed. A cell was considered active if it produced at least one action potential during the recording. This is now clearly stated in the Methods (page 42). As the reviewer pointed out correctly, we cannot provide the frequency of the APs as the recordings were variable in length. However, we have now included F-I curves for evoked APs (this point is addressed in more detail in a later response below).

Results/ Figure 1g

How many action potentials must a given neuron need to fire to be considered active? What was the criteria for an event to be counted as an action potential?

A cell was considered active if it generated at least one action potential during the recording. We added the information to the figure captions and Methods (page 42). Action potentials were identified using three main criteria: the signal had to exceed a minimum amplitude threshold, cross 0 mV to confirm a full reversal of the membrane potential, and be followed by a clear afterhyperpolarization. In our analysis script, these criteria were implemented with the following parameters: a minimum peak prominence of 20 mV, a minimum peak height of 0 mV, a maximum peak width of 60 ms, and a minimum inter-peak interval of 20 ms. Using these parameters, the automated action potential detection aligned perfectly well with manual identification. We have now added these information to the Methods section (page 43).

*Also, it would be great to display these data as **F-I curves** (frequency or number of action*

potentials fired during the 200ms step as a function of the amount of current injected into soma). This way, we can see how spiky the neurons are – a measure of intrinsic excitability – not just whether they are able to spike in response to somatic current injection.

As suggested by the reviewer, we now provide F-I curves, which indeed provide more detailed information on the intrinsic excitability of the neurons (new Figure 2 k). We have also conducted new experiments in which we patched Piezo1 KD and control neurons on soft and stiff substrates at DIV7. For these cells too, we provide the F-I curves for evoked APs as requested by the reviewer (new Figure 2o).

Results/ Figure 2

Next, the authors ask whether piezo1 underlies the effect of stiff substrate on neuronal activity. For this, the activity of piezo1 kd neurons on stiff and soft substrate is measured via somatic calcium imaging instead of measuring IE via whole cell electrophysiology. What is the rationale for switching from the electrophysiological approaches for measuring neural activity to calcium imaging? To improve logical flow of the experiments, perhaps the rationale for the different approaches for measuring neuronal activity (electrophys Fig. 1 versus calcium imaging Fig. 2) could be explained in the manuscript.

We thank the reviewer for this suggestion. While patch-clamp recordings provide much more detailed electrophysiological data, the low throughput would not have allowed to collect sufficient data across all six conditions at three different time points. Sticking with patch clamp electrophysiology would thus have rendered the project unfeasible within the duration of a postdoc position. Therefore, we switched to calcium imaging, as the relationship between calcium transients and action potentials is well-established and it enables the analysis of much larger data sets within a reasonable timeframe (please also see our answer to reviewer 1).

We have added the motivation for the change in methods to the manuscript (page 10). Additionally, the new patch clamp data set included in the revised version of our manuscript align very well with the original calcium imaging results as the activity measured with patch clamp shows the same trends as the calcium imaging, thus closing a potential gap between our patch-clamp and calcium imaging data.

Figure 2: The heading of Figure 2: Shouldn't this be "Piezo1 delays electrical maturation of neurons on stiff substrates"?

We thank the reviewer very much for catching this! We corrected the mistake as suggested.

Results/ Figure 3

In the section "Piezo1 regulates substrate stiffness-dependent GABAR and GLUR1 expression via Transthyretin":

Second paragraph, last sentence of this section states that Piezo1 KD led to similarly high TTR expression and electrical activity on soft and stiff substrates and references Fig. 3b. However, Figure 3B only shows TTR expression levels. It does not show electrical activity.

We have corrected this by also referring to Figure 2g (now Fig. 3g).

Figure 3f does show electrical activity of TTR KD on soft tissue versus Control neurons on stiff substrate, but it seems this panel is not referenced in the Results section. This needs

to be corrected.

We apologise for this omission. The figure panel is now referenced in the last sentence of the paragraph: "In agreement with an inhibitory role of Piezo1 activity on TTR expression levels, the fraction of active TTR KD cells cultured on soft substrates was as low as that of CTRL neurons cultured on stiff substrates (Fig. 4f), confirming a direct link between substrate stiffness, Piezo1 activity, TTR expression levels and neuronal maturation in vitro."

Figure 3c: (western blots of TTR, GluR1, GABAR, and NMDAR)

Fig. 3c (left): What is "TPS"? It is not mentioned in the legend nor in the Results section. Please define this at least in the legend and explain why it is included in this blot.

Fig3c (middle and left): Please define tubulin in the legend and why it is included in the western blot.

We appreciate the reviewer's comments, which highlighted the need for further explanation of WB bands and their purpose. "TPS" stands for "total protein stain". Both TPS and tubulin have been used for normalisation of the data, which we now specify in the figure caption. As we explain in the Methods, we conducted a titration assay to determine the most suitable normalisation protein for each protein of interest. Our results indicated that TTR signals were most stable when normalised to TPS, whereas all other protein signals showed the highest stability when normalised to tubulin (please also see reply to next comment). We have updated the figure caption to clarify the normalisation approach. Further details on the choice of normalisation staining are provided in the Methods section (page 47) and our reply to the next comment.

Fig. 3d: it would be comforting to see the signal ratio for tubulin included here, or at least mention its ratio in the legend (since it is probably a control).

We thank the reviewer for raising this point. We had assumed that tubulin levels would not be impacted by substrate stiffness. However, this is difficult to confirm experimentally as we do not have another housekeeping gene, which we could use for signal normalisation, whose expression is independent of substrate stiffness. Furthermore, total protein levels (TPS) were not suitable for normalising tubulin as tubulin signals normalised by TPS varied significantly with changes in lysate concentration (see Fig. R1 below).

Fig. R1: The ratio of tubulin : total protein (TPS) decreased considerably for increasing lysate concentrations. Hence, TPS could not be used for normalising the tubulin signal.

For the same reason, TPS could not be used to normalise the levels of synaptic receptors (see Fig. R2 below). As the ratio of receptor proteins to tubulin was fairly constant across different lysate concentrations, we had initially decided to normalise receptor levels by tubulin. However, because we cannot fully rule out the possibility that tubulin levels might vary between conditions, we have now removed all Western blot analyses from Fig. 4 that relied on the normalisation by tubulin. The key message of the figure – that TTR levels regulate the electrical maturation of neurons downstream of Piezo1 (and substrate stiffness) – is not impacted though, as we had already normalised TTR levels by TPS in

the initial submission. Furthermore, the regulation of synapses by substrate stiffness is already shown in the immunofluorescence images and plots in Fig. 1.

Fig. R2: Normalisation of synaptic receptor levels by tubulin levels was much more consistent if compared to TPS. The ratio of receptor proteins to tubulin was fairly constant across different lysate concentrations, while that of receptor proteins to TPS changed considerably. Hence, TPS could not be used for normalisation.

Discussion

The model (Figure 3e) does not include a proposed mechanism on how the stiff substrate ultimately impairs maturation of intrinsic excitability. Perhaps this could be acknowledged and/or a theory could be proposed. For instance, could the decrease in synaptic strength somehow hinder that maturation of IE? Alternatively, is it known whether TTR has a direct effect on Na⁺ channels?

This is an excellent question. While we are not aware of any direct effects of TTR on sodium channels, we have shown that in stiff environments synapse densities are reduced (Fig. 1). Changes in synapse densities, on the other hand, can trigger homeostatic adjustments in intrinsic excitability, thus linking synaptic and intrinsic plasticity (e.g., Zbili et al., PNAS 2021). We hypothesize that this way TTR indirectly links substrate mechanics to the maturation of the intrinsic excitability of neurons. We have included this hypothesis in the discussion (page 13-14).

Methods:

Electrophysiology

For the voltage-step protocol, the baseline voltage at which neurons were held at and brought back to after each voltage step needs to be indicated.

The baseline voltage was -60mV. We added this information to the Methods section (page 41).

Why was lucifer yellow included in the internal recording solution?

Lucifer Yellow was included for quality control after recording of the cell, to exclude leaking cells. This is now mentioned in the Methods sections (page 42).

*The Methods states that the neuron's membrane resistance was measured. However, membrane resistances do not appear to be reported. Given that the ability of a neuron to fire action potentials is shaped by its membrane resistance, the **average membrane resistances** for neurons on soft and stiff substrates is important to know and should be reported. The **resting membrane potentials** – an indicator of overall neuron health – should also be reported if possible.*

We have plotted these parameters and included them in Extended Data Figure 4.

We are very grateful for the reviewers' enthusiasm and appreciation for the work, and for the encouraging feedback and excellent questions. Addressing their suggestions has significantly improved our work. Thank you!

Response to Reviewers

We would like to thank the reviewers for their encouraging comments, and reviewer 2 for their excellent feedback and suggestions. We were able fully to address all of reviewer 2's remaining points as detailed below. Changes in the manuscript are highlighted in yellow.

Reviewer Comments and Answers to the Comments:

Reviewer #1 (Remarks to the Author):

The authors have done a good job of addressing my previous comments. In particular, the new data in Figure 5 very nicely addressed one of my central concerns. I have no further comments.

We thank the reviewer for their positive assessment of our work, and we would like to thank them again for the time and effort they spent helping us improve this manuscript.

Reviewer #2 (Remarks to the Author):

Reviewer's Comments:

Reviewer #2 (Remarks to the Author)

1. While the new manuscript has been improved with new experiments and clarification of statistical details, there is still some opaqueness that remains. For example, comparing the %active cells in figure 2 (electrophysiology) vs figure 3(ca imaging), there are differences in equivalent groups. In fig 2, in soft substrate at DIV7, the percent active is around 50%, while in figure 3, under the same conditions, it is much more lower (~20%).

We would like to thank the reviewer for acknowledging that our new manuscripts has been improved. The differences between the cells in Figs. 2 and 3 very likely arise from differences in the experimental conditions. The patch-clamp recordings shown in Figure 2a–k were performed on wild-type cells, whereas in old calcium imaging experiments shown in Figure 3 we used CRISPR control cells. As described in the Methods, these CRISPR control cells were electroporated post-isolation and transfected with a Cas9 complex fused with tracer RNA in order to meaningfully compare them to the cells in which Piezo was downregulated using CRISPR/Cas9. Hence, these control cells experienced more stress than the wild-type cells used in Fig. 2, which very likely explains the slight quantitative differences between these different cell types. However, we would like to point out that, qualitatively, these cells behaved very similarly, with much earlier spontaneous activity on soft substrates if compared to stiff ones. We have now clarified this point in the main text on page 10.

Likewise in the Ca⁺⁺ imaging figure, at 5DIV the soft substrate cells show a substantial amount of activity. The authors say the median value in this condition is 0, but this is deceptive. From what I can see, about 8/21 fields show significant activity, the median is zero because more fields showed no activity, but that's ignoring 1/3 of the data!

We now realised that our wording might have been misleading, we apologise. We changed the wording in the main text to: "At DIV 5, we detected very little activity across all conditions.". Similarly we changed the wording of the caption of Fig. 3k to: "CTRL neurons

(red) on soft substrates became active between DIV 5 and DIV 6, whereas those on stiff substrates showed little activity BY DIV 7.”.

Also how was cell health assessed in fields with zero activity? Without ground truth experiments, I am unconvinced the Ca⁺⁺ transients necessarily represent the same measure as the electrophysiology, and it concerns me that while some trends are the same, some are different.

Cell health was assessed by visual inspection. Cells that showed morphological alterations indicating pathological changes, such as blebbing, beading, or poor cell-to-substrate adherence, were not used for experiments. We added this clarification to the Methods (page 41).

We agree that calcium imaging is not providing the same measure as electrophysiology. Patch clamp electrophysiology provides more detailed information about individual cells and offers unique insights. This is why we followed the reviewer’s previous suggestion and measured Piezo1 KD and control cells with patch clamp in addition to the calcium imaging data that we had already shown in our initial submission. However, calcium imaging is a well-established method to assess neuronal activity, and all new (and old) patch clamp experiments showed the same trends between the different conditions as the calcium imaging experiments. There is not a single trend that is qualitatively different between the experiments. In both experimental approaches, neurons became active sooner on softer substrates, and their slowed down maturation on stiffer substrates was inhibited by knockdown of Piezo1. We feel that new experiments investigating which details of neuronal activity are similar between patch clamp and calcium imaging in our system would go beyond the scope of this study.

2. The added patch clamp studies in the Piezo mutants was a good addition, I would report the values of the ANOVA describes in the text, the area under the curve measurements for the FI curves are not standard in the field.

We want to thank the reviewer for acknowledging the improvement of the work. We have now changed the statistical analysis to ANOVA as suggested and updated the figures as well as the description in the main text (page 8) accordingly.

3. I am also confused by this comment in the rebuttal "We respectfully disagree with the reviewer. The role of TTR in regulating glutamate and GABA receptor expression has been established in multiple prior studies, which we now reference more explicitly in the revised manuscript. And, as acknowledged by the reviewer, we unambiguously show that substrate stiffness regulates TTR via Piezo1. Hence, our data strongly support a mechanistic, not purely correlative, relationship." There used to be data on GABA and Glu receptors, but that has been removed, so unclear how the authors continue so support that claim, or otherwise I must have missed their point.

We are sorry for any confusion caused by our response, which we had written before deciding to remove the receptor data. We forgot to update our response subsequently and apologise for this oversight. We agree with the reviewer that we do not show a direct link between substrate stiffness and synapse receptors, and have adjusted our discussion accordingly (page 14).

4. The authors claim to show in Sup Fig15 to show data where they "knocked down TTR expression in neurons and used calcium imaging to determine the fraction of active cells

on soft and stiff substrates at DIV 7", but these data are missing, the referenced figure shows tubular staining, which is also not clear what that was for.

We agree with the reviewer, Extended Data figure 15 just confirms the downregulation of TTR in our cells. To avoid confusion, we moved the figure reference in the text to an earlier position:

To confirm a role of TTR downstream of Piezo1, we knocked down TTR expression in neurons (Extended Data Fig. 15, Methods) and used calcium imaging to determine the fraction of active cells on soft and stiff substrates at DIV 7.

The calcium imaging data referred to in this sentence are shown in Fig. 4f, which is mentioned in the following paragraph.

In sum, there are improvements, but some statistical decisions remain opaque, the data is disorganized (eg. why is the Piezo KD data in figure 2 and not in figure 3 where it better fits)? The figures are hard to read with tiny labels (eg. Figure 2). Thus the paper is extremely hard to follow. I suggest an overhaul of figure design, with clearer stats and selection of which data are relevant and which aren't.

We thank the reviewer for suggesting to change the figure design, which indeed makes it much easier to read and digest in particular Fig. 2. We have now moved all Piezo1 KD data from figure 2 to figure 3 as suggested, increased the font sizes in the figure panels, and removed unnecessary figure panels.

Reviewer #3 (Remarks to the Author):

All concerns have been well-addressed. Love the new F-I curves! Great work!

We thank the reviewer for the positive assessment of our work and want to thank them again for the time and effort they spent helping us improve this manuscript.

Response to Reviewers

We would like to thank reviewer 2 for their feedback and last suggestions, which we addressed in the supplementary material.

Reviewer Comments and Answers to the Comments:

Reviewer #2 (Remarks to the Author):

The authors have addressed most of my concerns, the adjusted statistics and figure redesign have helped with clarity. I still think there's more going on with the Ca⁺⁺ imaging that the authors are overlooking, but I agree that at least the trends are all in the same direction, so we'll go with that.

I also noticed in Extended Figure 2 (sorry that I missed this earlier), with the Sholl analysis, that the authors perform some sort of "area under the curve" analysis again. This is not standard in the field (at least not as far as I know), unclear what the units (μm) even tell you since it is measuring intersections, and they should change this to an ANOVA or some other test to compare differences in branch distribution. If they want other quantitative comparisons they could for example compare total dendritic branch length, or total number of branch tips.

Other than that, I think the general conclusions of the paper stand.

We thank the reviewer for their careful evaluation of our manuscript and for acknowledging that the general conclusions of the paper are sound. We also appreciate their helpful comment regarding our use of the area-under-the-curve analysis in the Sholl plots. We agree that this metric is not the most intuitive for describing neuronal branching. While measures such as total dendritic branch length provide valuable information about overall growth, they do not capture branching complexity, which other reviewers specifically encouraged us to assess.

A standard approach in Sholl analysis to capture the branching behaviour is to count the intersections of all neurites per cell with each concentric Sholl ring, sum these intersections, and compare the total values across conditions. We agree that this measure is more intuitive than our previous analysis, and we have therefore revised our approach accordingly. The figure and the caption have been changed as described.

The changes in the caption read:

For each cell, all neurite–ring intersections were quantified and compared across conditions using a two-way ANOVA with two-sided Sidak's post hoc test. Branching behaviour was unaffected by substrate stiffness in both CTRL and KD cells.